# Deep Learning of Compositional Targets with Hierarchical Spectral Methods

Hugo Tabanelli [1]   Yatin Dandi [1,2]   Luca Pesce [1]   Florent Krzakala [1]

## Abstract

Why depth yields a genuine computational advantage over shallow methods remains a central open question in learning theory. We study this question in a controlled high-dimensional Gaussian setting, focusing on compositional target functions. We analyze their learnability using an explicit three-layer fitting model trained via layer-wise spectral estimators. Although the target is globally a high-degree polynomial, its compositional structure allows learning to proceed in stages: an intermediate representation reveals structure that is inaccessible at the input level. This reduces learning to simpler spectral estimation problems, well studied in the context of multi-index models, whereas any shallow estimator must resolve all components simultaneously. Our analysis relies on Gaussian universality, leading to sharp separations in sample complexity between two and three-layer learning strategies.

## 1. Introduction

Deep neural networks consistently outperform shallow models across a wide range of learning tasks, yet providing a rigorous and quantitative explanation for the computational advantages conferred by depth remains a central open problem in machine learning theory (Sejnowski, 2020; Zhang et al., 2021). While classical approximation theory establishes that deep architectures can represent certain functions more efficiently than shallow ones (Telgarsky, 2016; Mhaskar et al., 2017; Poggio et al., 2017), approximation alone does not explain why such functions should be *learnable* from data using feasible sample sizes. A fundamental theoretical challenge can thus be phrased as:

*Q1: Can one quantify the computational advantages of depth for learning structured targets in high dimensions?*

Several lines of work have provided partial answers in controlled settings. Deep *linear* networks admit an exact characterization of training dynamics and implicit bias (Saxe et al., 2014; Ji & Telgarsky, 2019; Arora et al., 2019; Lee et al., 2019; Ghorbani et al., 2021), but their limited expressivity prevents the emergence of genuinely hierarchical features. Another successful direction studies multi-index models, where the target depends on the input only through a fixed low-dimensional projection (Ben Arous et al., 2021; Ba et al., 2020; Ghorbani et al., 2020; Bietti et al., 2022; Abbe et al., 2023; Troiani et al., 2024). A large part of this literature is formulated in a teacher-student setting: labels are generated by a structured teacher function, and one asks whether a learning algorithm, or student model, can recover the relevant low-dimensional structure from samples. This viewpoint also extends to broader neural-network teacher-student models (Gardner & Derrida, 1989; Aubin et al., 2018; Buzaglo et al., 2025). In the present work, we adopt the same controlled teacher-student philosophy, but focus on a compositional teacher designed to isolate the role of depth. While mathematically rich, multi-index targets are often efficiently learnable by shallow architectures (Arnaboldi et al., 2024; Lee et al., 2024), limiting their usefulness for isolating a fundamental advantage of depth. This motivates a more structural question:

*Q2: What class of target functions can reveal a provable separation between shallow and deep learning strategies?*

There has been some recent progress in this direction. For instance, the study of *Random Hierarchy Models* (RHMs), introduced by Cagnetta & Wyart (2024) as a generative model for compositional data, provides a clean benchmark where empirical evidence suggests a dramatic separation between shallow and deep architectures in terms of sample complexity. An orthogonal line of work concerns the *computational advantage of depth through training dynamics* for structured, hierarchical target functions in high-dimensional Gaussian settings: Dandi et al. (2025) introduced a class of high-dimensional hierarchical target functions and showed that deep networks trained by gradient descent can learn them with drastically fewer samples than any shallow model. Closer to us, and a major inspiration to the present work,

[1] Information, Learning and Physics Laboratory, École polytechnique fédérale de Lausanne (EPFL), Lausanne, Switzerland [2] Statistical Physics of Computation Laboratory, École polytechnique fédérale de Lausanne (EPFL), Lausanne, Switzerland . Correspondence to: Hugo Tabanelli <hugo.tabanelli@epfl.ch>.

*Proceedings of the 43$^{rd}$ International Conference on Machine Learning*, Seoul, South Korea. PMLR 306, 2026. Copyright 2026 by the author(s).

is the series of papers (Nichani et al., 2024; Wang et al., 2023; Fu et al., 2025) that discuss compositional functions and demonstrated the advantage of three-layer nets over two layers and kernel methods.

In all these approaches, the key insight is that gradient-based training does not learn all features at once, but instead *progressively reveals structure across layers*, effectively decomposing a *complex learning task into a sequence of simpler ones*. This progressive evolution of features across layers is precisely what enables deeper architectures to tackle functions that remain inaccessible to shallow models.

In this work, we follow this perspective in the simplest tractable setting. We focus on a class of high-dimensional target functions that are globally complex, being high-degree polynomials of the input, but admit a *compositional structure* as a sequence of nonlinear polynomial transformations. While such targets can exhibit arbitrarily rich behavior, their compositional form allows them to be decomposed into a hierarchy of simpler intermediate features: early layers capture coarse structure, while later layers progressively refine and assemble higher-level representations.

Rather than analyzing gradient descent directly, which leads to technically delicate dynamics even in simplified models, we adopt a more transparent approach. We replace gradient-based training by an explicit sequence of simple, layer-wise *spectral estimators* that implement the same progressive decomposition extending prior works (Nichani et al., 2024; Fu et al., 2025; Dandi et al., 2025). This hierarchical spectral training framework isolates the computational role of depth independently of optimization dynamics, and allows us to study genuine multi-layer feature learning, with multiple successive steps of feature recovery.

Within this framework, we analyze how multi-layer hierarchical strategies confer a computational advantage by enabling a staged spectral *disentanglement* of compositional targets. Rather than learning the full mapping $f^\star = g^\star \circ h^\star$ in a single step, learning proceeds sequentially by identifying intermediate structure and reusing it across layers, providing a clean explanation for the advantage of depth in learning compositional structure (Sejnowski, 2020).

## 2. Settings, Main Results, and Related Work

**Motivation.** To motivate our class of targets, consider the following observation: For prediction tasks on real images or natural language, the initial layers of trained networks extract *low-degree* features such as edge detectors for images or $n$-gram clusters for sentences. This can be interpreted as the selection of a *sparse subspace* of low-degree functions of inputs. The subsequent layers again progressively filter non-linear features to eventually reach a label. Thus, one can model hierarchical data as an iterative composition of

blocks of the following kind:

$$
\begin{array}{ccccc}
\text{High-} & & \text{space of} & & \\
\text{dimensional} & \longrightarrow & \text{low-degree} & \longrightarrow & \text{sparse} \\
\text{inputs} & & \text{features} & & \text{selection}
\end{array}
\qquad (1)
$$

**Data and Target Functions.** We wish to make our high-dimensional setting as simple as possible to isolate the advantage of depth for learning compositional functions. We shall assume that we are given an i.i.d. dataset $\{(\mathbf{x}_\mu, y_\mu)\}_{\mu=1}^n$ with Gaussian inputs $\mathbf{x}_\mu \sim \mathcal{N}(0, I_d)$ and labels $y_\mu$ given by the following *compositional target* functions $y_\mu = f^\star(\mathbf{x}_\mu)$ as a general non-linearity on top of latent hierarchical polynomial features:

$$
\mathbf{x}_\mu \in \mathbb{R}^d \longrightarrow \mathbf{h}_\mu^{(1)} \in \mathbb{R}^{d^\epsilon} \longrightarrow h_\mu^{(2)} \in \mathbb{R}, \qquad (2)
$$

defined as

$$
h_i^{(1)} := \langle A_i^{(1)}, H_k(\mathbf{x}) \rangle, \quad i = 1, \dots, d^\epsilon = d_1, \qquad (3)
$$
$$
h^{(2)} := \langle A^{(2)}, H_2(\mathbf{h}^{(1)}) \rangle, \quad d_2 = 1 \qquad (4)
$$

and finally the label function is found by taking a general non-linearity (which we can assume to be a polynomial of degree $p$) $g^\star(\cdot)$ on top of the latest layer non-linear features:

$$
y_\mu = f^\star(\mathbf{x}_\mu) = g^\star(h_\mu^{(2)}) \qquad (5)
$$

The tensors $\{A_i^{(1)} \in \mathbb{R}^{d \times \cdots \times d}\}_{i=1}^{d^\epsilon}$ are symmetric with independent Gaussian entries of variance $\Theta(d^{-k})$ and have an effective number of parameters $D_1 = \mathcal{B}(d, k) = \binom{d+k-1}{k}$. The mapping $\mathbf{x} \to h_i^{(1)} := \langle A_i^{(1)}, H_k(\mathbf{x}) \rangle$ can be interpreted as a sparse selection of features in the space of degree-$k$ polynomials defined in $\mathbf{x}$. The matrix $A^{(2)} \in \mathbb{R}^{d_1 \times d_1}$ is symmetric with independent Gaussian entries of variance $\Theta(d_1^{-2})$.

We have used the order-$k$ Hermite tensors [1]. The bracket $\langle \cdot, \cdot \rangle$ denotes the Frobenius inner product $\langle U, V \rangle := \mathrm{Tr}(U^\top V)$. This reads for order-2 Hermite tensors

$$
\langle A, H_2(\mathbf{h}) \rangle = \frac{1}{\sqrt{2}} \left( \mathbf{h}^\top A \mathbf{h} - \mathrm{Tr}(A) \right) \qquad (6)
$$

With this definition, $f^\star$ becomes a high-degree polynomial in $\mathbf{x}$. We shall consider the problem in high-dimension, with

$$
d \to \infty, \qquad d_1 \propto d^\varepsilon, \qquad n \propto d^\alpha, \qquad (7)
$$

with exponents $\epsilon, \alpha \geq 0$ left free. The targets studied in (Nichani et al., 2024; Wang et al., 2023) are closely related models for hierarchical feature learning. For instance, (Nichani et al., 2024) considers targets of the form

---

[1]Formally defined by the tensorial Rodrigues formula $\sqrt{k!} H_k(\mathbf{x}) = (-1)^k e^{\|\mathbf{x}\|^2/2} \nabla^{\otimes k} \left( e^{-\|\mathbf{x}\|^2/2} \right)$ where $\nabla^{\otimes k}$ denotes the $k$-fold symmetric tensor of derivatives.

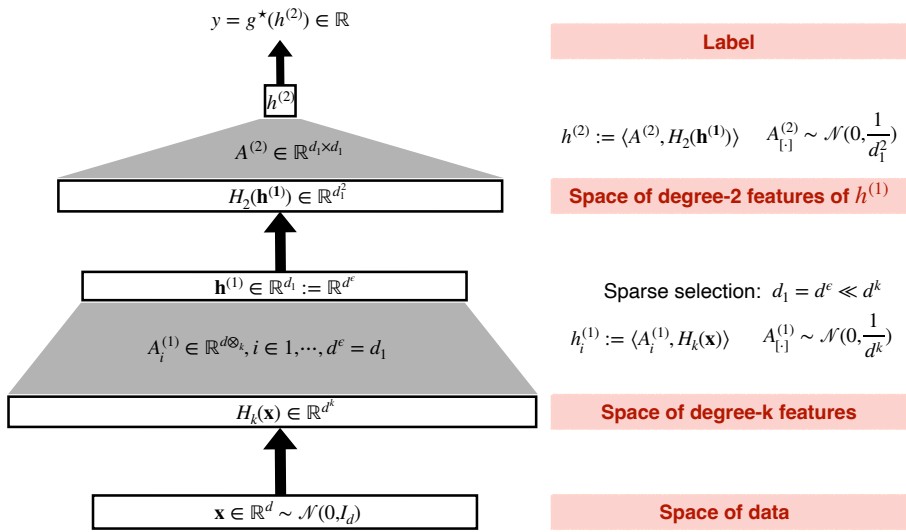

*Figure 1.* An illustration of the compositional target functions as defined in section 2.

$f^\star = g^\star(\mathbf{x}^\top A\mathbf{x}) = g^\star(\langle A, H_2(\mathbf{x})\rangle)$, which corresponds to the case $k = 2$, $d_1 = 1$ ($\epsilon = 0$), and $A^{(2)} = 1$ in our notation, while (Wang et al., 2023) considers the higher-degree case $k \geq 2$. Here we study a different high-dimensional regime, where the intermediate representation has a growing dimension $d_1 = d^\epsilon$. Perhaps the simplest example is given by the following "three-layer" target, where we only consider 2nd-order Hermite:

$$h_{\mu,i}^{(1)} := \langle A_i^{(1)}, H_2(\mathbf{x}_\mu)\rangle, \qquad i = 1, \ldots, d_1, \quad (8)$$

$$h_\mu^{(2)} := \langle A^{(2)}, H_2(\mathbf{h}_\mu^{(1)})\rangle \qquad d_2 = 1 \quad (9)$$

$$y_\mu = f^\star(\mathbf{x}_\mu) = g^\star(h_\mu^{(2)}). \quad (10)$$

In the simplest case, when $g^\star(.)$ is just the identity, $f^\star$ is a quartic function of $\mathbf{x}_\mu$ but has a "decomposable structure" $\mathbf{x}_\mu \rightarrow \mathbf{h}_\mu^{(1)} \rightarrow g^\star(h_\mu^{(2)})$ as a composition of two squares.

**Expected performance: heuristic.** Our goal is to efficiently learn the target function $f^\star(\mathbf{x})$ from the dataset $\mathcal{D} = \{(\mathbf{x}_\mu, y_\mu)\}_{\mu=1}^n$, and to understand how depth can improve sample complexity.

We begin by recalling a fundamental limitation of **kernel methods and random feature models** (Rahimi & Recht, 2007). Such methods can only learn a polynomial approximation of degree $\kappa$ in the Hermite expansion of $f^\star$ when the number of samples scales as $n = \mathcal{O}(d^\kappa)$ (Mei et al., 2022). As a consequence, these approaches are insensitive to the compositional structure of the target and depend only on its overall polynomial degree. For instance, in the simple example (10), if we use $g^\star(x) = x$, then $f^\star$ is quartic in $\mathbf{x}$, and kernel methods require $n = \mathcal{O}(d^4)$ samples. More generally, if the outer nonlinearity $g^\star$ is a polynomial of degree $p$, the required sample size scales as $\mathcal{O}(d^{4p})$.

In contrast, multi-layer strategies exploit the compositional structure of the target. In a related regime, (Fu et al., 2025) show that a three-layer network can learn $f^\star$ with $n = \mathcal{O}(d^4 + d^{\varepsilon p})$, already improving over shallow methods when the outer nonlinearity has low degree. While their setting differs from ours in the intermediate feature dimension and the upper-layer structure, this result points to the same general phenomenon: depth can make a globally high-degree target accessible by revealing and exploiting intermediate structure.

In our model, the growing intermediate layer suggests a sharper layer-wise counting heuristic. Learning the feature map from $\mathbf{x}$ to $\mathbf{h}^{(1)}$ requires on the order of $d^{k+\varepsilon}$ samples, corresponding to the number of parameters in the first feature layer. Once this representation is recovered, the second-layer quadratic form can be estimated in the latent coordinates with only $\mathcal{O}(d^{2\varepsilon})$ samples, after which fitting the final one-dimensional nonlinearity is trivial as the effective dimensionality has been reduced to one. When $k \geq 2$, this suggests that the overall sample complexity scales as

$$n = \mathcal{O}(d^{k+\varepsilon}),$$

which, for example, becomes $n = \mathcal{O}(d^{2+\varepsilon})$ when $k = 2$.

This is precisely the scaling achieved by our hierarchical approach. We construct a multi-layer learning procedure that first recovers the feature map from $\mathbf{x}$ to $\mathbf{h}^{(1)}$ using a spectral method, akin to Principal Component Analysis (PCA). Once this intermediate representation has been identified, the second-layer quadratic form can be estimated directly in the recovered latent coordinates, up to the natural rotation ambiguity of the first-layer features. More generally, the procedure can be viewed as a hierarchical spectral training scheme: instead of learning the full target in a single step,

it first uses degree and rank selection to reveal the relevant first-layer feature subspace, and then uses the recovered representation as the appropriate basis in which the upper-layer structure becomes a low-order estimation problem. This provides a tractable proxy for deep feature learning, inspired by the spectral estimators used in multi-index models (Damian et al., 2022; 2024), while remaining fully analyzable in the high-dimensional limit.

From this perspective, depth enables a progressive reparameterization of the data: after the first spectral step, the original high-degree problem is transformed into a quadratic estimation problem in the learned latent coordinates. This staged disentanglement of compositional structure is what allows deep architectures to succeed in regimes where shallow estimators confront the full high-order complexity of the target at once. Although our approach does not analyze gradient descent directly, it captures the same staged feature-learning mechanism observed in gradient-based training of deep networks, and one can establish explicit connections between gradient descent and the spectral estimators appearing in our method; see App. C.

**Summary of Main Results**

We summarize our main results below:

• **Hierarchical spectral recovery:** We introduce an explicit hierarchical spectral estimator that reconstructs the latent representations $\mathbf{h}^{(1)}$ and $h^{(2)}$ using Hermite moment matrices and PCA-type spectral methods, relying on classical eigenvalue separation phenomena (e.g. BBP transitions (Baik et al., 2005)). We characterize the regimes of dimension $(d, d^\varepsilon)$ and sample size $n$ under which each stage of the hierarchical procedure succeeds. The latent features $\mathbf{h}^{(1)} \in \mathbb{R}^{d^\varepsilon}$ can be consistently recovered provided that the number of samples satisfies

$$n \gg d^{k+\varepsilon}, \tag{11}$$

and that the signal subspace remains sufficiently sparse in the ambient degree-$k$ Hermite feature space, namely

$$d^\varepsilon \ll d^k. \tag{12}$$

Conditioned on the recovery of $\mathbf{h}^{(1)}$, the scalar second-layer feature $h^{(2)}$ can be estimated as soon as

$$n \gg d^{2\varepsilon}. \tag{13}$$

Fitting the final one-dimensional nonlinearity $g^\star$ then requires no additional computational complexity. For $k > 1$, the recovery of $\mathbf{h}^{(1)}$ dominates the overall complexity, yielding a total sample complexity

$$n_{\text{tot}} = \mathcal{O}(d^{k+\varepsilon}) \tag{14}$$

This leads to sharp separations between shallow and multi-layer learning strategies.

• **Numerical validation:** While our rigorous proof is restricted to $\varepsilon < 1/2$ and relies on the bounded operator norm of the estimator of the parameters, we provide extensive numerical experiments illustrating the multi-layer learning procedure and confirming the predicted sample-complexity scaling beyond these regimes. Moreover, despite the asymptotic nature of the theory ($d \to \infty$), the observed transitions occur already at moderate dimensions.

• **Gaussian Equivalence:** Our analysis relies on a principle of independent interest, that extends the Gaussian Equivalence Principle previously established for shallow models to hierarchical settings, asserting that at each layer, suitably normalized representations behave asymptotically as Gaussian vectors with explicitly computable covariances.

### 2.1. Further Related Work

**Random feature and kernel methods.** A large part of the theory of neural networks comes from regimes where features are effectively fixed during training, most notably kernel methods and random feature (RF) models (Rahimi & Recht, 2007). These approaches admit sharp generalization guarantees in high dimensions (Gerace et al., 2020; Goldt et al., 2022; Mei et al., 2022; Xiao et al., 2022; Defilippis et al., 2024), but their expressive power is fundamentally limited. In particular, kernel and RF methods can only recover low-degree polynomial approximations of the target, with an effective degree bounded by the number of samples and features (Mei et al., 2022).

**Multi-index and spectral methods.** Despite substantial progress in understanding fixed-feature methods, a central challenge in learning theory remains a principled description of how learning algorithms adapt to low-dimensional structure. A canonical framework to study this phenomenon is provided by *multi-index models*, where the target depends on the input only through a small number of linear projections followed by a nonlinear map. The information-theoretic limits of these models are well understood (Barbier et al., 2019; Aubin et al., 2018), and a large body of work has characterized their algorithmic learnability, highlighting intrinsic limitations of kernel methods (Mei et al., 2022) and the role of algorithmic thresholds and hardness exponents for neural networks (Ben Arous et al., 2021; Abbe et al., 2022; Ba et al., 2020; Dandi et al., 2024; Damian et al., 2024; Arnaboldi et al., 2024; Lee et al., 2024; Montanari & Wang, 2026). Recent results show that suitable algorithmic variants of SGD can achieve near-optimal sample complexity for this class (Arnaboldi et al., 2024; Lee et al., 2024; Troiani et al., 2024).

The spectral estimators we employ build directly on this line of work, in particular in (Damian et al., 2024). A variety of spectral methods for estimating low-dimensional structure in high-dimensional models have been developed

and analyzed in related contexts (Lu & Li, 2020; Mondelli & Montanari, 2018; Maillard et al., 2022). Spectral approaches tailored to multi-index models have been proposed and shown to achieve optimal or near-optimal performance (Kovačević et al., 2025; Defilippis et al., 2025). Our work extends these to a hierarchical setting, where spectral estimation is combined with rank selection and cleaning procedures.

**Hierarchical and compositional models.** Depth is often argued to be effective because it allows networks to exploit hierarchical or compositional structure in the data. In addition to the works discussed in the introduction, this intuition has motivated a variety of hierarchical target and data models, including, e.g., tree-structured functions and random hierarchy models (Mossel, 2016; Daniely, 2017; Poggio et al., 2017; Allen-Zhu & Li, 2019; Abbe et al., 2022; Ren et al., 2023; Garnier-Brun et al., 2025; Cagnetta & Wyart, 2024; Cagnetta et al., 2024). The idea that learning proceeds by extracting structure across successive scales is closely related to coarse-graining and renormalization concepts from statistical physics (Wilson, 1971). Such connections have inspired several theoretical studies of deep learning (Mehta & Schwab, 2014; Li & Wang, 2018; Marchand et al., 2023; Dandi et al., 2025).

**Gaussian Equivalence Principles.** A central theme underlying much recent progress—and a key technical enabler of the present work—is the emergence of *Gaussian equivalence* or *universality* principles, whereby the behavior of learning algorithms on non-Gaussian data can be characterized through an equivalent Gaussian model. Beginning with the seminal work of El Karoui (2008) on the spectrum of sample covariance matrices, such Gaussian universality properties have since been established in a wide range of learning settings. These include generalized linear estimation and random feature models (Gerace et al., 2020; Goldt et al., 2022; Mei & Montanari, 2022; Dhifallah & Lu, 2020; Vilucchio et al., 2025), empirical risk minimization and neural tangent kernel regimes (Montanari & Saeed, 2022), as well as mixtures of Gaussians (Dandi et al., 2023). Of particular relevance to our analysis are recent extensions of Gaussian equivalence to *quadratic* and higher-order polynomial feature models (Bandeira & Maillard, 2025; Xu et al., 2025; Wen et al., 2025; Xiao et al., 2022; Hu et al., 2024). Understanding the spectral behavior of kernel and moment matrices in these polynomial regimes has become a central topic in random matrix theory (Lu & Yau, 2025).

## 3. Hierarchical Spectral Methods

We now describe an explicit hierarchical spectral procedure for learning the compositional targets $f^\star(\mathbf{x})$. The algorithm recovers nonlinear features layer by layer through low-order

---

**Algorithm 1** Hierarchical spectral learning

**input** Data $\{(\mathbf{x}_\mu, y_\mu)\}_{\mu=1}^n$, max degree $K_{\max}$
1: **First layer recovery:**
2: **for** $k = 1$ to $K_{\max}$ **do**
3:    Compute $\phi_\mu^{(k)} = \mathcal{F}[H_k(\mathbf{x}_\mu)] \in \mathbb{R}^{D_k}$ and

$$\widehat{C}_k^{(1)} = \frac{1}{n} \sum_{\mu=1}^n y_\mu H_2\left(\phi_\mu^{(k)}\right) \in \mathbb{R}^{D_k \times D_k}.$$

4: **end for**
5: Select the degree $\widehat{k}_1$ as the smallest $k \leq K_{\max}$ for which $\widehat{C}_k^{(1)}$ exhibits a low-rank structure. Let $\widehat{d}_1$ be the corresponding rank.
6: Compute the top $\widehat{d}_1$ eigenvectors $\{\widehat{A}_i^{(1)}\}_{i=1}^{\widehat{d}_1}$ of $\widehat{C}_{\widehat{k}_1}^{(1)}$.
7: **for** $\mu = 1$ to $n$ and $i = 1$ to $\widehat{d}_1$ **do**
8:

$$\widehat{h}_{\mu,i}^{(1)} \leftarrow \left\langle \widehat{A}_i^{(1)}, \mathcal{F}[H_{\widehat{k}_1}(\mathbf{x}_\mu)] \right\rangle.$$

9: **end for**
10: **Second layer recovery:**
11:

$$\widehat{A}^{(2)} \leftarrow \widehat{C}_2^{(2)} = \frac{1}{n} \sum_{\mu=1}^n y_\mu H_2\left(\widehat{\mathbf{h}}_\mu^{(1)}\right) \in \mathbb{R}^{\widehat{d}_1 \times \widehat{d}_1}.$$

12: **for** $\mu = 1$ to $n$ **do**
13:

$$\widehat{h}_\mu^{(2)} \leftarrow \left\langle \widehat{A}^{(2)}, H_2\left(\widehat{\mathbf{h}}_\mu^{(1)}\right) \right\rangle.$$

14: **end for**
15: Perform kernel regression on $\{(\widehat{h}_\mu^{(2)}, y_\mu)\}_{\mu=1}^n$.
16: **Return** $\widehat{A}^{(1)}, \widehat{A}^{(2)}$

---

moment matrices and spectral thresholding. At each stage, it extracts the latent representations $\mathbf{h}^{(1)} \in \mathbb{R}^{d^\varepsilon}$ and $h^{(2)} \in \mathbb{R}$ defined in (3), using only second-order spectral information.

**Recovery of the first-layer features $\mathbf{h}^{(1)}$.** We begin by explaining how the algorithm recovers the first-layer latent features $\mathbf{h}^{(1)}$ from the input data.

Let $F[H_k(\mathbf{x})]$ denote a fixed linear flattening of the order-$k$ Hermite tensor, accounting for symmetry such that the Frobenius inner product is preserved. Therefore, $F[H_k(\mathbf{x})]$ represents a vector living in a space of dimension $\mathcal{B}(d, k) = \mathcal{O}(d^k)$, corresponding to the effective dimension of degree-$k$ multivariate Hermite polynomials. Our estimator constructs the empirical moment matrix

$$\widehat{C}_k^{(1)} := \frac{1}{n} \sum_{\mu=1}^n y_\mu H_2(F[H_k(\mathbf{x}_\mu)]), \qquad (15)$$

which can be viewed as a second-order covariance oper-

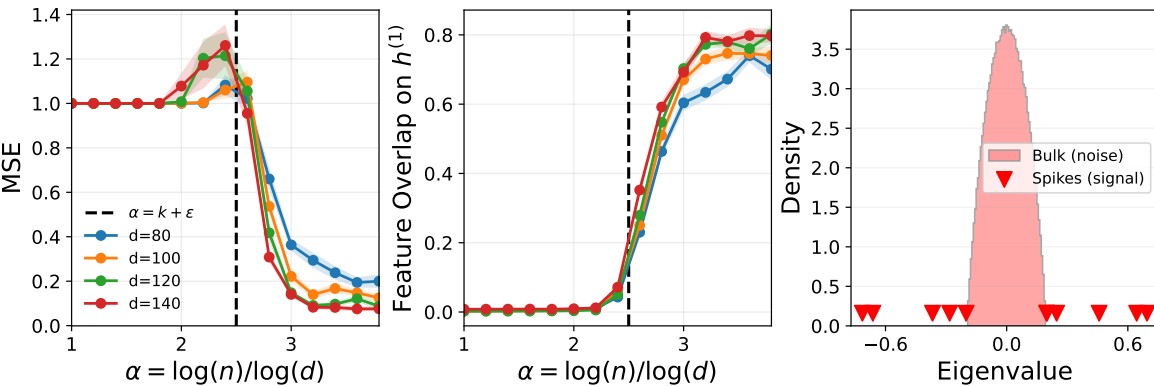

*Figure 2.* **Learning with hierarchical spectral methods:** This plot shows the performance of the hierarchical estimator described in Algorithm 1 when learning the target (10) with an identity readout $g^\star(x) = x$. In this case, kernel and shallow methods require $O(d^4)$ samples. **Left:** Mean Squared Error (MSE) achieved by the labels predictor $\{\hat{y}_\mu\}_{\mu=1}^n$ versus normalized number of samples $\alpha$ for different input dimensions $d = \{80, 100, 120, 140\}$. The latent features' dimension is fixed to $d^\epsilon = \sqrt{d}$. The MSE drops significantly at the theoretically predicted threshold $n = \mathcal{O}(d^{k+\epsilon}) = \mathcal{O}(d^{2.5})$ in agreement with Theorem 3.3. **Center:** Evaluation of the learned representations $\{\hat{h}_\mu^{(1)}\}_{\mu=1}^n$ measuring an overlap with the ground truth (Details in Appendix B). Similarly to the behaviour of the MSE, the overlap grows significantly at the predicted threshold $d^{2.5}$. **Right:** Spectrum of the second-order matrix $\hat{C}_2^{(1)}$ in eq. (25) for a fixed $d = 100$ and $\alpha = 3$. The density of eigenvalues presents a clear separation in a supported bulk (noise) plus separate $d^\epsilon = 10$ spikes (signal), separating from the bulk (noise), as predicted by the theory.

ator acting on this feature space. In the population limit, $\hat{C}_k^{(1)}$ exhibits a low-rank structure: its rank is $\mathcal{O}(d^\varepsilon)$, corresponding to the span of the tensors $\{A_i^{(1)}\}_{i=1}^{d^\varepsilon}$ defining the first-layer features. Under the scaling regime $n \gg d^k d^\varepsilon$ and $d^k \gg d^\varepsilon$ this signal subspace separates sharply from noise via a BBP-type spectral transition.

As a consequence, both the correct polynomial degree $k$ and the span of $\{A_i^{(1)}\}_{i=1}^{d^\varepsilon}$ can be recovered by simple eigenvalue thresholding. Our analysis relies on the vectors $F[H_k(\mathbf{x})]$ asymptotically behaving as Gaussian vectors in dimension $D_1$ which we detail in Section 4. Below we briefly explain how this equivalence translates to concrete predictions on the number of samples required towards the recovery of the parameters.

Consider the model with $F[H_k(\mathbf{x}_\mu)]$ replaced by $\tilde{\mathbf{x}} \sim \mathcal{N}(0, I_{d_k})$. For $\mu \in [n]$, let $\mathbf{x}_\mu^\star, \mathbf{x}_\mu^\perp$ denote the projection of $\mathbf{x}_\mu$ on the subspace spanned by $\{A_i^{(1)}\}_{i=1}^{d^\varepsilon}$ and its orthogonal complement respectively. With a slight abuse of notation, we denote by $A^{(1)} \in \mathbb{R}^{d_1 \times D_1}$ the matrix defined by stacking the rows obtained by flattening the tensors $\{A_i^{(1)}\}_{i=1}^{d^\varepsilon}$. Under Assumption 3.1, the rows of $A^{(1)}$ are asymptotically orthonormal, $A^{(1)}A^{(1)\top} = I_{d_1} + o(1)$ in operator norm in the regime considered. In the equivalent model, the labels $y_\mu$ then depend on $\tilde{\mathbf{x}}$ only through projections onto the matrix $A^{(1)}$, i.e., through $\mathbf{x}_\mu^\star$.

By the independence of $\mathbf{x}_\mu^\star, \mathbf{x}_\mu^\perp$, we obtain the following signal + noise decomposition of $\hat{C}_k^{(1)}$, analogous to the decomposition of spectral estimators for single, multi-index

models (Lu & Li, 2020; Mondelli & Montanari, 2018; Defilippis et al., 2025; Kovačević et al., 2025):

$$\hat{C}_k^{(1)} \simeq \mathbb{E}[\hat{C}] + \mathbb{V}[\hat{C}] \tag{16}$$

$$\mathbb{E}[\hat{C}_k^{(1)}] = \text{Signal} = \nu_1 A^{(1)\top} A^{(2)} A^{(1)} \tag{17}$$

$$\mathbb{V}[\hat{C}_k^{(1)}] = \text{Noise} = \frac{1}{n}\tilde{X}_\perp Y \tilde{X}_\perp^\top + o_d(1), \tag{18}$$

where $\nu_1$ denotes the first Hermite coefficient of $g^\star$. Recall that, by assumption we have $d_1 \ll d^k$. Hence the signal component $S^\star := \nu_1 A^{(1)\top} A^{(2)} A^{(1)}$ appears as $d_1$ spikes in the $d_k \times d_k$ matrix $\hat{C}_k^{(1)}$.

Moreover, under the Gaussian equivalent model, the entries of $\tilde{X}_\perp$ are independent of the labels $y_\mu$ and hence the matrix $\frac{1}{n}\tilde{X}_\perp Y \tilde{X}_\perp^\top$ corresponds to an isotropic bulk.

Our main result then shows that for $d, d_1 \to \infty$ with $d_1 \ll d$ and $n \gg d^k d_1$, the overlaps of the estimator $\hat{C}$ along $A_1^{(1)}, \cdots, A_{d_1}^{(1)}$ converge to the corresponding overlaps of the "signal" matrix $S^\star := \nu_1 A^{(1)\top} A^{(2)} A^{(1)}$, given by $A^{(1)} S^\star A^{(1)\top} = \nu_1 A^{(2)}$.

Throughout, our results rely on the following assumptions:

**Assumption 3.1.** For $i = 1, \cdots, d_1$, $A_i^{(1)}$ are symmetric with independent entries $\sim \mathcal{N}(0, d^{-k})$

**Assumption 3.2.** $g^\star : \mathbb{R} \to \mathbb{R}$ is uniformly Lipschitz and satisfies $\mathbb{E}_{z \sim \mathcal{N}(0,1)}[g^\star(z)z] \neq 0$.

The theorem below is a consequence of the Gaussian equivalence discussed in Sec. 4 and matrix concentration results:

**Theorem 3.3.** *Let $\hat{C}_k^{(1)}$ be as defined in Eq.* (15). *Then, whp as $d, d_1 \to \infty$:*

$$\sqrt{d_1} \left\| A^{(1)} \hat{C}_k^{(1)} A^{(1)\top} - \nu_1 A^{(2)} \right\|_2$$
$$= \tilde{O}\left( \sqrt{\frac{d^k d_1}{n}} \right) + \tilde{O}\left( \frac{d_1}{\sqrt{d}} \right) + \tilde{O}\left( \sqrt{\frac{1}{d_1}} \right), \quad (19)$$

*where $\tilde{O}$ includes polylogarithmic factors. The $\sqrt{d_1}$ scaling accounts for the normalization $\left\| A^{(2)} \right\|_2 = \mathcal{O}(\frac{1}{\sqrt{d_1}})$.*

The proof of the theorem is provided in Appendix A.

*Remark* 3.4 (**Conjectured Extension**). While the above result only applies to the overlaps of $\hat{C}_k^{(1)}$ along $A^{(1)}$, one can more generally show that the estimator $\hat{C}_k^{(1)}$ and the signal matrix $S^\star := \nu_1 A^{(1)\top} A^{(2)} A^{(1)}$ are asymptotically equivalent in terms of projections onto "generic isotropic tensors", that do not possess non-vanishing contractions. We detail this generalized equivalence and the class of such tensors in Appendix A.1, which includes random tensors $\{A_i^{(1)}\}_{i=1}^{d_1}$ with high probability. We conjecture that the top eigenvectors of $\hat{C}_k^{(1)}$ obey such genericity and hence converge to the top eigenvectors of $S^\star$. Under such a conjecture, the subspace of $A^{(1)}$ can be recovered by thresholding $\hat{C}_k^{(1)}$ upto its top $d_1$ eigenvalues (Algorithm 1).

**Estimation of the second-layer feature $h^{(2)}$.** Once the signal subspace associated with the first layer has been recovered (let $\hat{d}_1$ be the dimension of this subspace), yielding estimates for $\{\hat{A}_i^{(1)}\}_{i=1}^{\hat{d}_1}$, the latent features are estimated by

$$\widehat{h}_{\mu,i}^{(1)} = \left\langle \widehat{A}_i^{(1)}, H_k(\mathbf{x}_\mu) \right\rangle, \qquad i = 1, \ldots, \widehat{d}_1, \quad (20)$$

yielding the reconstructed representation $\widehat{\mathbf{h}}_\mu^{(1)} \in \mathbb{R}^{\widehat{d}_1}$. Conditioned on this reconstruction, the estimation of the second-layer feature $h^{(2)}$ becomes a quadratic estimation problem in the recovered latent coordinates. The algorithm forms the empirical moment matrix

$$\widehat{C}_2^{(2)} := \frac{1}{n} \sum_{\mu=1}^{n} y_\mu \, H_2\left( \widehat{\mathbf{h}}_\mu^{(1)} \right), \quad (21)$$

which acts on the order-2 Hermite feature space associated with $\widehat{\mathbf{h}}^{(1)}$. In the population limit: this matrix aligns with the matrix $A^{(2)} \in \mathbb{R}^{d^\varepsilon \times d^\varepsilon}$ which directly estimates the second-layer quadratic form, up to the rotation ambiguity inherited from the first-layer recovery. In the idealized setting where $\widehat{\mathbf{h}}_\mu^{(1)}$ is replaced by the true features $\mathbf{h}_\mu^{(1)}$, this matrix converges to $\nu_1 A^{(2)}$ in operator norm, as soon as $n \gg d^{2\epsilon}$, as stated below. Consequently, once the first-layer representation has been identified, recovering $h^{(2)}$ only requires estimating this quadratic form in dimension $d_1 = d^\epsilon$.

Analogous to Theorem 3.3, for sufficiently large $d$, and small enough $d_1$, the vectors $h_\mu^{(1)}$ behave as vectors with jointly independent Gaussian entries in $\mathbb{R}^{d_1}$. Consequently we obtain that the matrix $\widehat{C}_2^{(2)}$ converges to the order-2 Hermite matrix of $y$ as a function of $\mathbf{h}_\mu^{(1)}$ asymptotically given by $\nu_1^\star A^{(2)}$ (See Lemma 4.2 for discussion).

**Theorem 3.5.** *Consider the idealized estimator $\tilde{C}_2^{(2)} := \frac{1}{n} \sum_{\mu=1}^{n} y_\mu \, H_2\left( \mathbf{h}_\mu^{(1)} \right)$ obtained by replacing $\widehat{\mathbf{h}}_\mu^{(1)}$ with the true features $\mathbf{h}_\mu^{(1)}$. We have w.h.p as $d, d_1 \to \infty$:*

$$\sqrt{d_1} \| \tilde{C}_2^{(2)} - \nu_1 A^{(2)} \|_2$$
$$= \tilde{O}\left( \frac{d_1}{\sqrt{d}} \right) + \tilde{O}\left( \sqrt{\frac{d_1^2}{n}} \right) + \tilde{O}\left( \sqrt{\frac{1}{d_1}} \right) \quad (22)$$

While Theorem 3.5 directly utilizes the true features $\mathbf{h}_\mu^{(1)}$, we expect the error bounds to hold for the estimates $\hat{\mathbf{h}}_\mu^{(1)}$ based on the conjectured equivalence in remark 3.4.

**Fitting of $g^\star$.** Once the latent features $\hat{h}^{(2)}$ are recovered, learning the function $g^\star$ amounts to perform a one-dimensional regression problem on $\{\hat{h}_\mu^{(2)}, y_\mu\}_{\mu=1}^n$, yielding the estimate for the labels $\{\hat{y}_\mu\}_{\mu=1}^n$ with standard one-dimensional kernel regression. This step does not affect the overall sample complexity scaling as it requires a number of samples independent of $d$ (e.g., (Vapnik, 1998)).

The complete algorithmic routine, including degree and rank selection via an elbow method, is summarized in Algo. 1.

**Examples and illustrations.** We illustrate the hierarchical spectral procedure on the simplest three-layer target defined in (10),

$$y_\mu = g^\star(h_\mu^{(2)}) = g^\star(\langle A^{(2)}, H_2(\mathbf{h}_\mu^{(1)}) \rangle) \quad (23)$$
$$h_{\mu,i}^{(1)} = \langle A_i^{(1)}, H_2(\mathbf{x}_\mu) \rangle, \quad (24)$$

In the identity case ($g^\star(h) = h$) the target $y_\mu$ is quartic in $\mathbf{x}_\mu$, and estimating it with a single-shot spectral method amounts to generic quartic regression. Such approaches naturally rely on fourth-order Hermite tensors and require $n = \mathcal{O}(d^4)$ data. In contrast, the hierarchical spectral estimator proceeds in two quadratic steps. At the first stage, it forms the moment matrix

$$\widehat{C}_2^{(1)} = \frac{1}{n} \sum_{\mu=1}^{n} y_\mu \, H_2\big( F[H_2(\mathbf{x}_\mu)] \big), \quad (25)$$

which acts on the $\mathcal{O}(d^2)$-dimensional space of symmetric matrices. In the population limit, $\widehat{C}_2^{(1)}$ has rank $\mathcal{O}(d^\varepsilon)$, with eigenvectors spanning $\text{span}\{A_i^{(1)}\}_{i=1}^{d^\varepsilon}$. From Theorem 3.3, we have that —provided $n \gg d^2 d^\varepsilon$ — this signal subspace

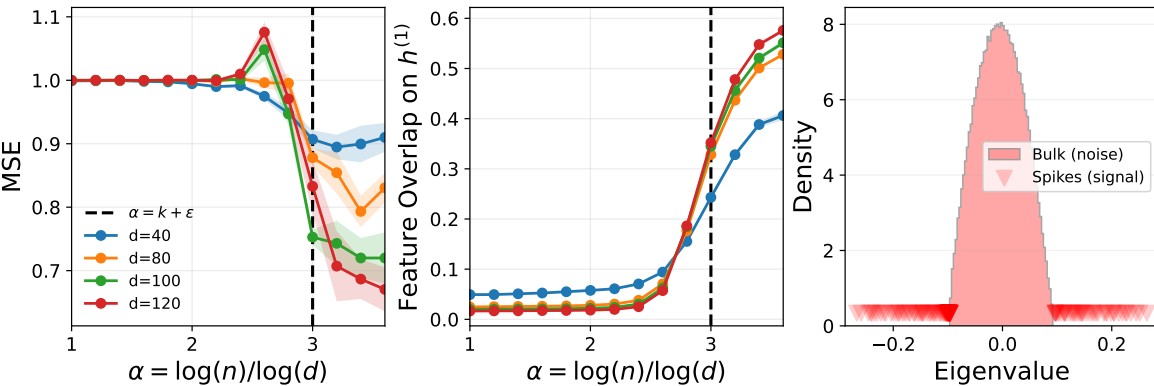

*Figure 3.* **On the role of $d^\epsilon$:** The plot shows the performance of the hierarchical estimator described in Algorithm 1 when learning a modification of target (10) seen in Fig. 2. We consider $\epsilon = 1$, therefore, an amount of spikes to learn equal to the ambient dimension $d$. Mean Squared Error (MSE) and Feature overlap are plotted versus normalized number of samples $\alpha$ for different input dimensions $d = \{40, 80, 100, 120\}$. Spectrum size $d = 120$.

separates from noise through a BBP-type transition (see the right panel in Fig. 2), allowing recovery of $\{A_i^{(1)}\}$ by spectral thresholding and yielding reconstructed features

$$\widehat{h}_{\mu,i}^{(1)} = \langle \widehat{A}_i^{(1)}, H_2(\mathbf{x}_\mu) \rangle.$$

Conditioned on this reconstruction, the problem reduces to a second quadratic estimation task in the latent space: estimating $A^{(2)}$ from the processed dataset $\mathcal{D}_1 = \{(\widehat{\mathbf{h}}_\mu^{(1)}, y_\mu)\}$. By Theorem 3.5, this step requires only $n \gg d^{2\varepsilon}$. As a result, the overall sample complexity scales as

$$n = \mathcal{O}\big(\max\{d^{2+\epsilon}, d^{2\varepsilon}\}\big),$$

which is dramatically smaller than the shallow quartic baseline $\mathcal{O}(d^4)$ whenever $d^\varepsilon \ll d^2$. In particular, when $d_1 = d^\varepsilon$ (e.g. $\varepsilon = 1/2$), this predicts a transition at $\alpha = 2 + \varepsilon$ (e.g. $\alpha = 2.5$), which is precisely what is observed in the numerical experiments in Fig. 2. Note the high value of the MSE at the interpolation peak ($n = \#(\text{parameters})$) corresponding to the characteristic double descent behavior (Belkin et al., 2019; Mei et al., 2022).

Few remarks are in order:

- The numerical illustrations show that the theoretical predictions given in Theorems 3.3, 3.5 for our hierarchical spectral procedure are valid well beyond the rate for $\varepsilon$ allowed by the rigorous scheme, i.e., $\varepsilon < \frac{1}{2}$. Fig. 2 considers $d^\epsilon = \sqrt{d}$ and we push this observation to the extreme setting where there are as many first-layer latent features as the ambient dimension, i.e., $\epsilon = 1$ (See Fig. 3).
- The experiments verify that the top-$d_1$ eigenvectors of $\widehat{C}_2^{(1)}$ lie along the subspace spanned by $A^{(1)}$, thus supporting the conjectured equivalence in Remark 3.4.
- Agreement with theoretical predictions is observed, even at moderate sizes, for general target functions (see with

$g^\star = \tanh$ in Fig. 4), or higher order Hermite (we refer to Appendix B for these additional numerical experiments).

## 4. Gaussian Equivalence

Theorems 3.3, 3.5 rely on a notion of asymptotic equivalence between the tensors $H_k(\mathbf{x})$ and Gaussian vectors in the corresponding dimension $D_1 = \binom{d+k-1}{k}$. In this section, we formalize the corresponding equivalence and sketch how it implies the results. Notions of asymptotic independence for low-degree polynomial functions of Gaussian variables are well established in the literature on Wiener chaos. In particular, central limit theorems for Wiener chaos imply that, for fixed $k$, linear projections of Hermite tensors of the form $\langle A, H_k(\mathbf{x}) \rangle$ converge in distribution to Gaussian random variables as the dimension grows (Nualart & Peccati, 2005; Nourdin & Peccati, 2009) whenever the tensors $A \in \mathbb{R}^{d \otimes_k d}$ have all non-trivial contractions vanishing.

More generally, many spectral and risk quantities associated with polynomial regression models are known to be asymptotically universal with respect to the precise distribution of the polynomial features (Hu et al., 2024; Xu et al., 2025; Wen et al., 2025). To prove Theorems 3.3, 3.5, it suffices to have such an equivalence for projections along fixed tensors as in the central limit theorems mentioned above.

**One-dimensional CLT.** The Lemma below is a direct consequence of the central limit theorem for Wiener Chaos (Nualart & Peccati, 2005; Nourdin & Peccati, 2009) (Theorem 4.1 in (Wen et al., 2025))

**Lemma 4.1.** *For any $\kappa, c > 0$ and bounded Lipschitz $\psi :$ $\mathbb{R} \to \mathbb{R}$, we have:*

$$\sup_{\substack{T : \|T \otimes_r T\|_F \le \frac{\kappa}{d^c} \\ \forall r \in [k-1]}} \mathbb{E}[\psi(\langle T, H_k(\mathbf{x}) \rangle)] - \mathbb{E}[\psi(\langle F(T), \tilde{\mathbf{x}} \rangle)] \xrightarrow[d \to \infty]{P} 0,$$

$$\tag{26}$$

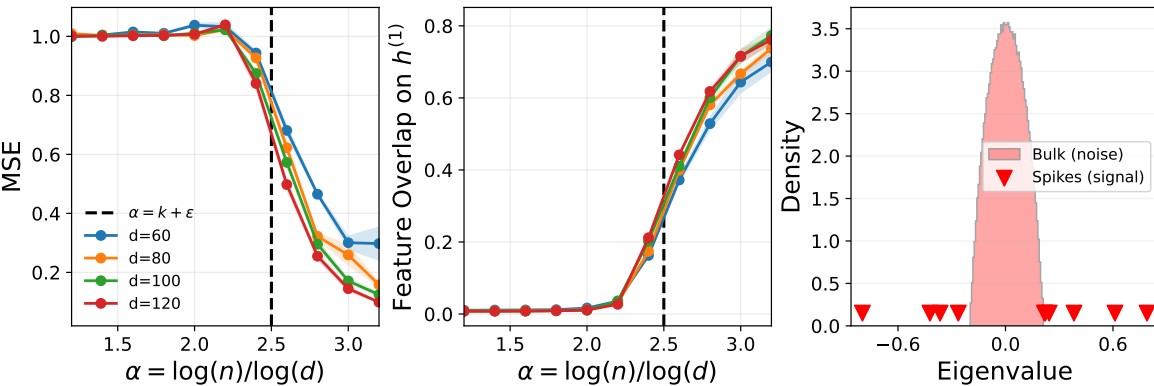

*Figure 4.* **On the role of $g^\star$.** Performance of the hierarchical estimator described in Algorithm 1 when learning a modified version of the target (10), as in Fig. 2. We consider the nonlinearity $g^\star = \mathtt{tanh}$. Introducing this additional nonlinearity does not alter the qualitative behavior of the method: once the first-layer features $\mathbf{h}^{(1)}$ are learned, estimating $g^\star$ reduces to a one-dimensional regression problem (see Algorithm 1). The mean squared error (MSE) and feature overlap are shown as functions of the normalized sample size $\alpha$, for input dimensions $d \in \{60, 80, 100, 120\}$. Spectrum size $d = 120$.

*where $\tilde{\mathbf{x}} \sim \mathcal{N}(0, I_{D_1})$ and $F(T)$ denotes the flattened version of the tensor $T$ and $T \otimes_r T$ denotes the symmetrized contraction of order $r$.*

Lemma 4.1 ensures that the features $H_k(\mathbf{x})$ behave as independent Gaussian vectors $\tilde{\mathbf{x}}$. It extends to the joint law of $r$ features $\langle A_1, H_k(\mathbf{x}) \rangle, \cdots, \langle A_r, H_k(\mathbf{x}) \rangle$ with the error in the joint law scaling as $\frac{r}{\sqrt{d}}$. We refer to Appendix A for the general non-asymptotic result derived through the approach in (Nourdin et al., 2010).

While Lemma 4.1 implies the asymptotic normality of the projections along tensors $H_k(\mathbf{x})$, they do not clarify the propagation of signal from $g^\star$ to $h^{(2)}(\mathbf{x})$ to $h^{(1)}(\mathbf{x})$. Specifically, the estimator $\hat{C}_k^{(1)}$ relies on the second-order Hermite components of $y$ as a function of $h^{(1)}$ lying in the span of $A^{(1)}$. This is a consequence of the following *composition of Hermite* lemma:

**Lemma 4.2** (Composition of Hermite features). *For $f \in L_2(\gamma)$, let $P_2(f)$ denote the coefficient matrix of the projection of $f$ onto Hermite polynomials of order 2. Let $\mathbf{u} \in \mathbb{R}^d$ and symmetric $A \in \mathbb{R}^{d \times d}$ with $\|A\|_F^2 = 1 + \tilde{O}_d(d^{-1/2})$, $\|A\|_2 = \mathcal{O}(\frac{1}{\sqrt{d}})$. For $g^\star$ satisfying Assumption 3.2, consider the quadratic feature $p(\mathbf{u}) = \langle A, H_2(\mathbf{u}) \rangle$ and the mapping $\mathcal{F}(\mathbf{u}) := g^\star(p(\mathbf{u}))$. We have:*

$$\sqrt{d}\, \|P_2(\mathcal{F}) - \nu_1 A\|_2 = \tilde{O}_d\left(\frac{1}{\sqrt{d}}\right) \qquad (27)$$

The proof is provided in Appendix A.6.

Lemma 4.2 combined with Lemma 4.1 ensures that $y_\mu$ asymptotically contains Hermite components of order 2 along $h^{(1)}(\mathbf{x})$ given by $\nu_1 A^{(2)}$. This in turn ensures that the expectation of $\hat{C}_k^{(1)}$ converges to the projection of these components onto the subspace $A^{(1)}$, resulting in the form

$A^{(1)\top} A^{(2)} A^{(1)}$ in Theorem 3.3.

## 5. Discussion and Limitations

We introduced a controlled high-dimensional model in which depth yields a sharp sample-complexity advantage for learning compositional targets. The key mechanism is that an intermediate representation turns a globally high-degree regression problem into a sequence of lower-order spectral estimation tasks.

The price paid for analytic tractability is that the setting remains structured. We assume Gaussian inputs and a teacher-student setting that is partially co-designed. A natural next step is to relax this feature-design aspect by replacing explicit Hermite coordinates with sufficiently rich nonlinear random features, while keeping the same layer-wise principle of label-weighted spectral selection.

This direction connects to the line of work on hierarchical polynomial learning (Nichani et al., 2024; Wang et al., 2023; Fu et al., 2025) and to recent work on Neural Low-Degree Filtering (Dandi et al., 2026), which develops random-feature spectral filtering as a more agnostic surrogate for hierarchical feature learning.

Several theoretical questions remain open. Our proof is restricted to $\epsilon < 1/2$, although the numerical experiments suggest that the predicted transition persists beyond this range. Extending the analysis to deeper hierarchies, more general nonlinearities, power law settings and a fully rigorous connection with gradient-based training are natural directions for future work.

# Acknowledgments

The authors would like to thank Joan Bruna, Alex Damian, Jason Lee, Yue Lu, Theodor Misiakiewicz and Lenka Zdeborova for helpful discussion and feedback. We acknowledge funding from the Swiss National Science Foundation grants OperaGOST (grant number 200021 200390), DSGIANGO (grant number 225837), and from the Simons Collaboration on the Physics of Learning and Neural Computation via the Simons Foundation grant (#1257412).

# Impact Statement

This paper presents work whose goal is to advance the field of Machine Learning. There are many potential societal consequences of our work, none of which we feel must be specifically highlighted here.

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

# A. Derivation of theoretical claims

## A.1. Tensors with vanishing contractions

For two symmetric tensors $S \in \mathbb{R}^{\otimes k}, T \in \mathbb{R}^{\otimes \ell}$ the symmetric contraction of order $r$ is defined as :

$$(S \otimes_r T)_{i_1 \dots i_{k-r} \, j_1 \dots j_{\ell-r}} = \sum_{a_1, \dots, a_r = 1}^{d} S_{i_1 \dots i_{k-r} \, a_1 \dots a_r} \, T_{j_1 \dots j_{\ell-r} \, a_1 \dots a_r}.$$

The condition on $T$ in Lemma 4.1 then states that the Frobenius norm of all non-trivial contractions i.e contractions of order $1, \cdots, k-1$ vanish as $d \to \infty$. For $k = 2$, the condition is equivalent to a bound on the operator norm of the matrix. It is easy to verify that the condition is satisfied for typical "isotropic" tensors such as tensors with independent Gaussian entries or tensors sampled uniformly over a fixed Frobenius norm (Lemma 5 in (Wen et al., 2025)). We require analogous conditions for the joint normality of multiple projections $\langle A_i, H_k(x) \rangle, \cdots, \langle A_r, H_k(x) \rangle$. As we discuss next, this is again ensured through bounds on all cross-contractions over pairs of $A_i, \cdots, A_r$

## A.2. Non-asymptotic joint-CLT for Wiener Chaos

Recall the definition the 1-Wasserstein distance (Kantorovich-Rubinstein) on $\mathbb{R}^r$ associated with the Euclidean norm:

$$d_W(G, Z) := \sup \left\{ \left| \mathbb{E}[h(G)] - \mathbb{E}[h(Z)] \right| : h : \mathbb{R}^r \to \mathbb{R}, \ \mathrm{Lip}(h) \leq 1 \right\}.$$

**Lemma A.1.** *For $r \in \mathbb{N}$, let $A_1, \cdots, A_i \in \mathbb{R}^{\otimes k}$ be independent symmetric tensors of order $k$ with i.i.d entries distributed as $\mathcal{N}(0, d^{-k})$. Let $Z \sim \mathcal{N}(0, I_r)$. There exists a constant $C_k < \infty$, depending only on $k$, such that for every $d \geq 2$,*

$$d_W([\langle A_i, H_k(x) \rangle, \cdots, \langle A_r, H_k(x) \rangle], Z) \leq C_k \frac{r}{\sqrt{d}}.$$

*Proof.* The proof follows through Corollary 3.6 in (Nourdin et al., 2010), which bounds the Wasserstein distance between joint distributions of vectors of the form $\langle A_i, H_k(x) \rangle, \cdots, \langle A_r, H_k(x) \rangle$ in terms of inner-products between their Malliavin derivatives.

We recall the following Lemma from (Nualart & Peccati, 2005), the Malliavin derivatives for functionals $\langle A_i, H_k(x) \rangle, \cdots, \langle A_r, H_k(x) \rangle$ are related to the contractions between pairs of tensors:

**Lemma A.2** (Lemma 2 in (Nualart & Peccati, 2005)). *For any $k, \ell \geq 1$, $S \in (\mathbb{R}^d)^{\odot k}$, and $T \in (\mathbb{R}^d)^{\odot \ell}$,*

$$\left( D\langle S, H_k(\mathbf{x}) \rangle \right)^{\top} \left( D\langle T, H_\ell(\mathbf{x}) \rangle \right) = k\ell \sum_{r=1}^{\min(k,\ell)} (r-1)! \binom{k-1}{r-1} \binom{\ell-1}{r-1} \langle H_{k+\ell-2r}(\mathbf{x}), (S \otimes_r T) \rangle,$$

*where $D$ denotes the Malliavin derivative.*

Hence, it suffices to bound the Frobenius norm of the contractions of matrices $\{A_i\}_{i=1}^r$. This is shown in the following Lemma:

**Lemma A.3.** . *Let $A, B$ be independent tensors in $(\mathbb{R}^d)^{\otimes k}$ with i.i.d. entries $\mathcal{N}(0, d^{-k})$. Then for each $s \in \{1, \dots, k\}$,*

$$\mathbb{E}\|A \otimes_s B\|_{\mathrm{F}}^2 = \Theta(d^{-s}).$$

*While for the self-contractions, $E\|A \otimes_s A\|_{\mathrm{F}}^2 = \Theta(d^{-s})$ for $s \in \{1, \dots, k-1\}$ and $E\|A \otimes_k A\|_{\mathrm{F}}^2 = 1$.*

*Proof.* Fix $s \in \{1, \dots, k-1\}$. For each pair of free multi-indices $(a, b) \in [d]^{k-s} \times [d]^{k-s}$,

$$(A \otimes_s B)_{a,b} = \sum_{m \in [d]^s} A_{a,m} B_{b,m}.$$

Since $A$ and $B$ have independent centered coordinates

$$\mathbb{E}\big[(A \otimes_s B)^2_{a,b}\big] = \sum_{m,n \in [d]^s} \mathbb{E}[A_{a,m} A_{a,n}] \, \mathbb{E}[B_{b,m} B_{b,n}]$$

$$= \sum_{m \in [d]^s} \mathrm{Var}(A_{a,m}) \, \mathrm{Var}(B_{b,m}) = d^s \cdot d^{-k} \cdot d^{-k} = d^{-(2k-s)}.$$

There are $\Theta(d^{2(k-s)})$ such pairs $(a, b)$, hence

$$\mathbb{E}\|A \otimes_s B\|_{\mathrm{F}}^2 = \sum_{a,b} \mathbb{E}\big[(A \otimes_s B)^2_{a,b}\big] = d^{2(k-s)} \cdot d^{-(2k-s)} = d^{-s}.$$

Analogously, we obtain the corresponding scaling for self-contractions.

$\square$

By substituting the above scaling in Lemma A.2, we obtain that the terms $\big(D\langle A_i, H_k(\mathbf{x})\rangle\big)^{\top} \big(D\langle A_j, H_\ell(\mathbf{x})\rangle\big)$ scale as $\mathcal{O}(\frac{1}{d})$ when $i \neq j$ and $1 - \mathcal{O}(\frac{1}{d})$ for $i = j$. Since the number of pairs $i, j$ is $\mathcal{O}(r^2)$, applying Corollary 3.6 in (Nourdin et al., 2010) gives the $\frac{r}{\sqrt{d}}$ bound in Lemma A.1. $\square$

### A.3. Tail bounds on non-linear features

In what follows, we will require the following control over the norms of $H_k(\mathbf{x})$:

**Lemma A.4** (Lemma F.4 in (Wen et al., 2025)). $\exists$ *constants* $c, C, C' > 0$ *such that with probability* $1 - Ce^{-cd}$:

$$\|F[H_k(\mathbf{x}_\mu)]\|^2 \leq C' d^k \tag{28}$$

### A.4. Proof of Theorem 3.3

We first consider the variance component. Write

$$\phi_\mu = F[H_k(\mathbf{x}_\mu)] \in \mathbb{R}^{D_1}, \qquad D_1 = \mathcal{B}(d, k) = \mathcal{O}(d^k).$$

The matrix $\widehat{C}_k^{(1)}$ is an average of matrices of the form $y_\mu H_2(\phi_\mu)$, whose leading term is the rank-one matrix $y_\mu \phi_\mu \phi_\mu^{\top}$. We apply Matrix Bernstein only after truncating these rank-one components, as in the proof of Lemma F.5 in (Wen et al., 2025). Let

$$\mathcal{E}_\mu = \Big\{ \|\phi_\mu\|^2 \leq C_0 D_1, \; |y_\mu| \leq C_y \Big\},$$

where the constants may absorb polylogarithmic factors. Lemma A.4, together with the polynomial moment bounds on the label, gives $\mathbb{P}(\mathcal{E}_\mu^c) \leq d^{-K}$ for any fixed $K$ upon absorbing by polylogarithmic factors in $C_0, C_y$. Define the truncated summand

$$Z_\mu = y_\mu H_2(\phi_\mu) \mathbf{1}_{\mathcal{E}_\mu}, \qquad X_\mu = \frac{1}{n} \left( Z_\mu - \mathbb{E} Z_\mu \right).$$

Conditioned on $\mathcal{E}_\mu$, we have

$$\|Z_\mu\|_2 \leq |y_\mu| \left( \|\phi_\mu\|^2 + 1 \right) = \tilde{\mathcal{O}}(D_1),$$

and hence the centered summands satisfy

$$\|X_\mu\|_2 \leq \tilde{\mathcal{O}}\left( \frac{D_1}{n} \right) = \tilde{\mathcal{O}}\left( \frac{d^k}{n} \right).$$

Furthermore, since $\mathbb{E}[\phi_\mu \phi_\mu^{\top}] = I_{D_1}$, we have:

$$\left\| \sum_{\mu=1}^{n} \mathbb{E} X_\mu^2 \right\|_2 \leq \frac{1}{n} \left\| \mathbb{E} Z_\mu^2 \right\|_2$$

$$\leq \frac{1}{n} \tilde{\mathcal{O}}\left( D_1 \left\| \mathbb{E} \phi_\mu \phi_\mu^{\top} \right\|_2 + 1 \right) = \tilde{\mathcal{O}}\left( \frac{D_1}{n} \right) = \tilde{\mathcal{O}}\left( \frac{d^k}{n} \right).$$

Matrix Bernstein applied to the independent centered matrices thus $\{X_\mu\}_{\mu=1}^n$ gives

$$\left\| \frac{1}{n} \sum_{\mu=1}^n Z_\mu - \mathbb{E} Z_\mu \right\|_2 = \tilde{\mathcal{O}} \left( \sqrt{\frac{d^k}{n}} + \frac{d^k}{n} \right) = \tilde{\mathcal{O}} \left( \sqrt{\frac{d^k}{n}} \right),$$

where the last step uses the regime $n \gtrsim d^k$. Since $\cap_{\mu=1}^n \mathcal{E}_\mu$ holds with probability greater than $1 - d^{-k}$ and the difference between $\mathbb{E} Z_\mu$ and the untruncated expectation is negligible by the same moment bounds, we obtain

$$\left\| \widehat{C}_k^{(1)} - \mathbb{E}[\widehat{C}_k^{(1)}] \right\|_2 = \tilde{\mathcal{O}} \left( \sqrt{\frac{d^k}{n}} \right) \tag{29}$$

Next we move on to the expectation. Let $\widehat{C}_g^{(1)}$ denote the corresponding estimator for the equivalent model with $F[H_k(\mathbf{x}_\mu)]$ replaced by standard Gaussian vectors $\tilde{\mathbf{x}} \sim \mathcal{N}(0, I_{D_1})$ i.e. $\widehat{C}_g^{(1)} := \frac{1}{n} \sum_{\mu=1}^n y_\mu H_2(\tilde{\mathbf{x}}_\mu)$. By the definition of $\hat{C}^{(1)}$ and $\hat{C}_g^{(1)}$, we have:

$$\langle A_i^{(1)} (\mathbb{E}[\hat{C}^{(1)} - \hat{C}_g^{(1)}]) A_j^{(1)} \rangle = \mathbb{E}[\langle A_i^{(1)}, H_k(\mathbf{x}) \rangle y(A^{(1)} F(H_k(\mathbf{x}))) \langle A_j^{(1)}, H_k(\mathbf{x}) \rangle] - \mathbb{E}[(\langle A_i^{(1)} \tilde{\mathbf{x}} \rangle) y(A^{(1)} \tilde{\mathbf{x}}) \langle A_j^{(1)} \tilde{\mathbf{x}} \rangle)]$$

We first reduce the operator norm to scalar projections. Let

$$M = A^{(1)} \mathbb{E}[\hat{C} - \hat{C}_g] A^{(1)\top},$$

Since $M$ is symmetric, the variational characterization gives

$$\sqrt{d_1} \|M\|_2 = \sqrt{d_1} \sup_{\|\mathbf{v}\|=1} \left| \mathbf{v}^\top M \mathbf{v} \right|.$$

It is therefore enough to control $\left| \mathbf{v}^\top M \mathbf{v} \right|$ uniformly over unit vectors $\mathbf{v}$. For such a $\mathbf{v}$, set

$$u = h^{(1)}(\mathbf{x}) = \left( \langle A_1^{(1)}, H_k(\mathbf{x}) \rangle, \ldots, \langle A_{d_1}^{(1)}, H_k(\mathbf{x}) \rangle \right)$$

and let $\tilde{u} = A^{(1)} \tilde{\mathbf{x}}$ denote the corresponding Gaussian feature vector. The quadratic form $\mathbf{v}^\top M \mathbf{v}$ is the difference between the true and Gaussian expectations of a scalar test function of $u$, with the relevant family of test functions being:

$$\Psi_{\mathbf{v}}(u) = \mathrm{He}_2(\mathbf{v}^\top u) \, g^\star \left( \langle A^{(2)}, H_2(u) \rangle \right).$$

Equivalently,

$$\left| \mathbf{v}^\top M \mathbf{v} \right| = \left| \mathbb{E} \Psi_{\mathbf{v}}(u) - \mathbb{E} \Psi_{\mathbf{v}}(\tilde{u}) \right|.$$

Thus, to apply Lemma A.1, we need a Lipschitz bound for the functions inside the expectations that is uniform over the projection direction $\mathbf{v}$. Fix $R = (\log d)^K$ with $K$ chosen large enough. For each fixed unit vector $\mathbf{v}$, define the cutoff event

$$\mathcal{E}_1(\mathbf{v}) := \left\{ |\mathbf{v}^\top u| \le R, \left| \langle A^{(2)}, H_2(u) \rangle \right| \le R, \|u\| \le R \sqrt{d_1} \right\},$$

On $\mathcal{E}_1(\mathbf{v})$, we have that $\mathrm{He}_2(\mathbf{v}^\top u)$ and its derivative are bounded by $\tilde{\mathcal{O}}(1)$ uniformly in $\mathbf{v}$. Moreover

$$\nabla_u \langle A^{(2)}, H_2(u) \rangle = \sqrt{2} A^{(2)} u, \qquad \left\| \sqrt{2} A^{(2)} u \right\| \le C \left\| A^{(2)} \right\|_2 \|u\| = \tilde{\mathcal{O}}(1),$$

using $\left\| A^{(2)} \right\|_2 = \mathcal{O}(d_1^{-1/2})$. Assumption 3.2 then gives that conditioned on $\mathcal{E}_1(\mathbf{v})$:

$$\sup_{\|\mathbf{v}\|=1} \mathrm{Lip}(\Psi_{\mathbf{v}}) = \tilde{\mathcal{O}}(1).$$

It remains to control the complement of the event $\mathcal{E}_1(\mathbf{v})$. Note that by Gaussian hypercontractivity, the bounds $\left| \langle A^{(2)}, H_2(u) \rangle \right| \le R, \|u\| \le R \sqrt{d_1}$ hold with probability greater than $1 - d^{-K}$ for any $K > 0$, as in Appendix F.2 and A.3 of (Wen et al., 2025).

Similarly,

$$Q(u) = \langle A^{(2)}, H_2(u) \rangle,$$

is a degree-$2k$ chaos with second moment $\mathcal{O}(1)$ under $\left\| A^{(2)} \right\|_F = \mathcal{O}(1)$ and $\left\| A^{(2)} \right\|_2 = \mathcal{O}(d_1^{-1/2})$. $Q(u)$ is thus bounded again by Gaussian hypercontractivity, as in Appendix F.2 and A.3 of (Wen et al., 2025). Consequently, for any $C_0 > 0$, choosing $K$ large enough in $R = (\log d)^K$ yields

$$\mathbb{P}(\mathcal{E}_1(\mathbf{v})^c) \leq d^{-C_0}$$

for every fixed unit $\mathbf{v}$, with constants independent of $\mathbf{v}$. On $\mathcal{E}_1(\mathbf{v})^c$, Holder's inequality gives

$$\mathbb{E}\big[|\Psi_{\mathbf{v}}(u)| \, \mathbf{1}_{\mathcal{E}_1(\mathbf{v})^c}\big] \leq d^{-C_0/2}$$

after increasing $C_0$ if necessary.

Consequently the joint CLT bound of Lemma A.1, applied to the $d_1$ variables defining $h^{(1)}$, can be used uniformly over the variational family $\{\Psi_{\mathbf{v}} : \|\mathbf{v}\| = 1\}$. This gives

$$\sqrt{d_1} \left\| A^{(1)} \mathbb{E}[\hat{C} - \hat{C}_g] A^{(1)T} \right\|_2 = \tilde{\mathcal{O}}(\frac{d_1}{\sqrt{d}}) \tag{30}$$

Finally, the proof is completely by noting that Lemma 4.2 implies that:

$$\sqrt{d_1} \left\| \mathbb{E}[\hat{C}_g] - \nu_1 (A^{(1)})^\top A^{(2)} (A^{(1)}) \right\| = \mathcal{O}(\frac{1}{\sqrt{d_1}}) \tag{31}$$

### A.5. Proof of Theorem 3.5

Similar to the proof of Theorem 3.3, we first consider the noise:

$$\tilde{C}_2^{(2)} - \mathbb{E}[\tilde{C}_2^{(2)}]. \tag{32}$$

Assuming $d_1 \ll d$, by Lemma A.4 and the joint moment bounds for the first-layer features, we have that $\left\| h^{(1)}(\mathbf{x}) \right\| \leq C\sqrt{d_1}$ w.h.p as $d \to \infty$. Truncating to this event, and to the corresponding polylogarithmic bound on $|y|$, the summands $y_\mu H_2(h^{(1)}(\mathbf{x}_\mu))$ have operator norm $\tilde{\mathcal{O}}(d_1)$. Their centered and normalized versions are therefore bounded by $\tilde{\mathcal{O}}(d_1/n)$. As above,

$$H_2(u)^2 \preceq C(\|u\|^2 uu^\top + I), \qquad \mathbb{E}[h^{(1)}(\mathbf{x}) h^{(1)}(\mathbf{x})^\top] \simeq I_{d_1},$$

Applying Matrix Bernstein to these truncated rank-one components yields

$$\left\| \tilde{C}_2^{(2)} - \mathbb{E}[\tilde{C}_2^{(2)}] \right\| = \tilde{\mathcal{O}}\left(\sqrt{\frac{d_1}{n}}\right) \tag{33}$$

Let $\tilde{C}_g^{(2)}$ denote the estimator obtained by replacing $h^{(1)}(\mathbf{x})$ by independent Gaussian entries $\tilde{\mathbf{x}} \sim \mathcal{N}(0, I_{d_1})$. We have:

$$\left\| \mathbb{E}[\tilde{C}_2^{(2)}] - \mathbb{E}[\tilde{C}_g^{(2)}] \right\|_2 = \sup_{\mathbf{v}:\|\mathbf{v}\|=1} \left| \mathbf{v}^\top \mathbb{E}[\tilde{C}_2^{(2)}] \mathbf{v} - \mathbf{v}^\top \mathbb{E}[\tilde{C}_g^{(2)}] \mathbf{v} \right|$$

Let $\mathbf{v} \in \mathbb{R}^{d_1}$ be arbitrary with $\|\mathbf{v}\| = 1$. Note that:

$$\mathbf{v}^\top \mathbb{E}[\tilde{C}_2^{(2)}] \mathbf{v} = \mathbb{E}[((\mathbf{v}^\top h^{(1)}(\mathbf{x}))^2 - 1) y(h^{(1)}(\mathbf{x}))] \tag{34}$$

Hence, we require control over expectations of test functions of the form:

$$\Psi_{\mathbf{v}}(u) = \mathrm{He}_2(\mathbf{v}^\top u) \, g^\star\Big(\langle A^{(2)}, H_2(u) \rangle\Big), \qquad \|\mathbf{v}\| = 1.$$

The same cutoff argument used in the proof of Theorem 3.3 earlier gives a Lipschitz bound uniform in $\mathbf{v}$. Indeed, for every fixed $\mathbf{v}$, $\mathbf{v}^\top h^{(1)}(\mathbf{x}) = \mathcal{O}(1)$ with high probability, conditioning on $|\mathbf{v}^\top h^{(1)}(\mathbf{x})| \leq R$, $\|h^{(1)}(\mathbf{x})\| \leq C\sqrt{d_1}$, and $|\langle A^{(2)}, H_2(h^{(1)}(\mathbf{x}))\rangle| \leq R$, both $\mathrm{He}_2(\mathbf{v}^\top u)$ and its derivative are $\tilde{\mathcal{O}}(1)$, while

$$\left\|\nabla_u \langle A^{(2)}, H_2(u)\rangle\right\| \leq C \left\|A^{(2)}\right\|_2 \|u\| = \mathcal{O}(1).$$

Assumption 3.2 therefore yields a $\tilde{\mathcal{O}}(1)$ Lipschitz constant for the truncated $\Psi_\mathbf{v}$, uniformly over $\|\mathbf{v}\| = 1$. The complement of this cutoff event is controlled by Gaussian-hypercontractivity as in the proof of Theorem 3.3. Hence, applying Lemma A.1 gives:

$$\sqrt{d_1} \left\|\mathbb{E}[\tilde{C}_2^{(2)}] - \mathbb{E}[\tilde{C}_g^{(2)}]\right\|_2 = \sqrt{d_1} \sup_{\mathbf{v}:\|\mathbf{v}\|=1} \left|\mathbf{v}^\top \mathbb{E}[\tilde{C}_2^{(2)}]\mathbf{v} - \mathbf{v}^\top \mathbb{E}[\tilde{C}_g^{(2)}]\mathbf{v}\right| = \tilde{\mathcal{O}}\left(\frac{d_1}{\sqrt{d}}\right) \tag{35}$$

Lastly, we note that Lemma 4.2 implies that $\sqrt{d_1} \left\|\mathbb{E}[\tilde{C}_g^{(2)}] - \nu_1 A^{(2)}\right\| = \tilde{\mathcal{O}}\left(\frac{1}{\sqrt{d_1}}\right)$. Combining with Equations 33, 35 then completes the proof.

### A.6. Proof of Lemma 4.2

Let

$$p(z) = \langle A, H_2(z)\rangle, \qquad B = P_2(\mathcal{F}),$$

so that the degree-2 projection of $\mathcal{F}$ is $\langle B, H_2(z)\rangle$. Since the degree-2 Hermite tensors are orthonormal for the Frobenius inner product, for every symmetric matrix $M$,

$$\langle B, M\rangle = \mathbb{E}\left[g^\star(p(z))\langle M, H_2(z)\rangle\right]. \tag{36}$$

It therefore suffices to prove that, uniformly over all unit vectors $\mathbf{v}$,

$$\left|\mathbb{E}\left[g^\star(p(z))\langle \mathbf{v}\mathbf{v}^\top, H_2(z)\rangle\right] - \nu_1\langle A, \mathbf{v}\mathbf{v}^\top\rangle\right| = \tilde{\mathcal{O}}\left(\frac{1}{d}\right), \tag{37}$$

because for the symmetric matrix $B - \nu_1 A$,

$$\|B - \nu_1 A\|_2 = \sup_{\|\mathbf{v}\|=1} \left|\langle B - \nu_1 A, \mathbf{v}\mathbf{v}^\top\rangle\right|.$$

Recall that by Gaussian integration-by-parts, for any twice differentiable function $\varphi : \mathbb{R}^d \to \mathbb{R}$ with polynomial growth and any symmetric matrix $M$,

$$\mathbb{E}\left[\varphi(z)\langle M, H_2(z)\rangle\right] = \frac{1}{\sqrt{2}} \mathbb{E}\left[\mathrm{Tr}\left(M\nabla^2\varphi(z)\right)\right]. \tag{38}$$

Indeed, the one-dimensional Gaussian integration-by-parts formula gives $\mathbb{E}[\varphi(z)(z_i z_j - \delta_{ij})] = \mathbb{E}[\partial_{ij}\varphi(z)]$ for every $i, j$, and summing against $M_{ij}/\sqrt{2}$ yields (38).

Apply (38) to $\varphi(z) = g^\star(p(z))$. Since

$$\nabla p(z) = \sqrt{2}Az, \qquad \nabla^2 p(z) = \sqrt{2}A,$$

we have

$$\nabla^2(g^\star \circ p)(z) = g^{\star\prime\prime}(p(z))\, \nabla p(z)\nabla p(z)^\top + g^{\star\prime}(p(z))\, \nabla^2 p(z).$$

Taking $M = \mathbf{v}\mathbf{v}^\top$ in (38) gives the exact identity

$$\mathbb{E}\left[g^\star(p(z))\langle \mathbf{v}\mathbf{v}^\top, H_2(z)\rangle\right] = \mathbb{E}\left[g^{\star\prime}(p(z))\right]\langle A, \mathbf{v}\mathbf{v}^\top\rangle + \sqrt{2}\,\mathbb{E}\left[g^{\star\prime\prime}(p(z))(\mathbf{v}^\top Az)^2\right]. \tag{39}$$

Thus the leading term is the desired direction $\langle A, \mathbf{v}\mathbf{v}^\top\rangle$, up to replacing $\mathbb{E}[g^{\star\prime}(p(z))]$ by $\nu_1$.

We now compare the scalar random variable $p(z)$ with a standard Gaussian. Diagonalize $A = \sum_{i=1}^d a_i w_i w_i^\top$, set $\psi_i(z) = \mathrm{He}_2(w_i^\top z)$, and write

$$p(z) = \sum_{i=1}^d a_i \psi_i(z).$$

The variables $\psi_i$ are independent, centered, variance-one random variables with finite moments of every fixed order. Moreover

$$\max_i |a_i| = \|A\|_2 = \mathcal{O}(d^{-1/2}), \qquad \sum_{i=1}^d a_i^2 = \|A\|_F^2 = 1 + \tilde{\mathcal{O}}(d^{-1/2}).$$

For every fixed polynomial $r$, this implies

$$\left| \mathbb{E}[r(p(z))] - \mathbb{E}_{G \sim \mathcal{N}(0,1)}[r(G)] \right| = \tilde{\mathcal{O}}(d^{-1/2}). \tag{40}$$

To see this, it suffices by linearity to take $r(t) = t^m$ with fixed $m$. Expanding $p(z)^m$ gives

$$\mathbb{E}[p(z)^m] = \sum_{i_1, \dots, i_m} a_{i_1} \cdots a_{i_m} \, \mathbb{E}[\psi_{i_1} \cdots \psi_{i_m}].$$

Applying (40) to $r = g^{\star\prime}$ and using the one-dimensional Stein identity,

$$\nu_1 = \mathbb{E}_{G \sim \mathcal{N}(0,1)}[g^\star(G)G] = \mathbb{E}_{G \sim \mathcal{N}(0,1)}[g^{\star\prime}(G)],$$

we get

$$\left| \mathbb{E}\left[ g^{\star\prime}(p(z)) \right] - \nu_1 \right| = \tilde{\mathcal{O}}(d^{-1/2}). \tag{41}$$

Since $\|A\|_2 = \mathcal{O}(d^{-1/2})$, the first error term in (39) is therefore bounded uniformly over $\|\mathbf{v}\| = 1$ by

$$\left| \mathbb{E}\left[ g^{\star\prime}(p(z)) \right] - \nu_1 \right| \left| \langle A, \mathbf{v}\mathbf{v}^\top \rangle \right| \le \tilde{\mathcal{O}}(d^{-1/2}) \, \|A\|_2 = \tilde{\mathcal{O}}\left( \frac{1}{d} \right).$$

It remains to control the second term in (39). Since $g^\star$ is a fixed polynomial, $g^{\star\prime\prime}$ is a fixed polynomial, and (40) applied to $r = (g^{\star\prime\prime})^2$ gives $\mathbb{E}[g^{\star\prime\prime}(p(z))^2] = \mathcal{O}(1)$. By Cauchy-Schwarz,

$$\left| \mathbb{E}\left[ g^{\star\prime\prime}(p(z))(\mathbf{v}^\top Az)^2 \right] \right| \le \left( \mathbb{E}[g^{\star\prime\prime}(p(z))^2] \right)^{1/2} \left( \mathbb{E}[(\mathbf{v}^\top Az)^4] \right)^{1/2}.$$

The scalar $\mathbf{v}^\top Az$ is Gaussian with variance $\|A\mathbf{v}\|^2$, hence

$$\left( \mathbb{E}[(\mathbf{v}^\top Az)^4] \right)^{1/2} = \sqrt{3} \, \|A\mathbf{v}\|^2 \le \sqrt{3} \, \|A\|_2^2 = \mathcal{O}\left( \frac{1}{d} \right).$$

Therefore the second term in (39) is also $\mathcal{O}(d^{-1})$, uniformly over $\|\mathbf{v}\| = 1$.

Combining the two estimates in (39) proves (37). Taking the supremum over unit vectors $v$ gives

$$\|P_2(\mathcal{F}) - \nu_1 A\|_2 = \tilde{\mathcal{O}}\left( \frac{1}{d} \right),$$

and multiplying by $\sqrt{d}$ yields the claimed $\tilde{O}_d(d^{-1/2})$ bound.

## B. Additional numerical experiments

In this section, we complement the main text with additional numerical experiments to validate the theoretical claims given in the main text.

**Computing the eigenvectors overlap.** While Algorithm 1 defines the estimators and the training procedure, it remains to specify the testing protocol. Let us assume that the training has been completed and that we have obtained the estimates $\{\widehat{A}_i^{(1)}\}_{i=1}^{\hat{d}^\epsilon}$ and $\widehat{A}^{(2)}$ of $\{A_i^{(1)}\}_{i=1}^{d^\epsilon}$ and $A^{(2)}$, respectively.

From Eq. (17), one can observe that $\widehat{A}^{(1)}$ recovers $A^{(1)}$ up to a rotation $R \in \mathbb{R}^{d_1 \times d_1}$, such that, at the population level,

$$\widehat{A}^{(1)} = R \, A^{(1)}. \tag{42}$$

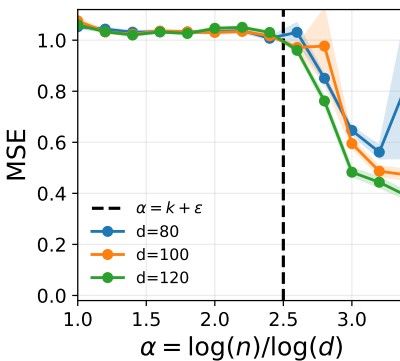 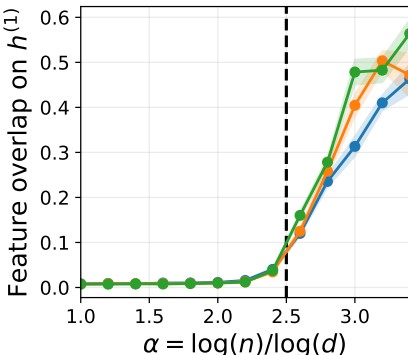

*Figure 5.* **Learning with a centered ReLU non-linearity.** Performance of the hierarchical estimator described in Algorithm 1 when learning the target (10) with $g^\star$ chosen as a centered ReLU function. **Left:** Mean Squared Error (MSE) of the label predictor $\{\hat{y}_\mu\}_{\mu=1}^n$ as a function of the normalized number of samples $\alpha = \log(n)/\log(d)$, for different input dimensions $d = \{80, 100, 120\}$. The latent feature dimension is fixed to $d^\varepsilon = \sqrt{d}$. Despite the non-polynomial nature of $g^\star$, the MSE exhibits a sharp drop around the theoretically predicted scaling $\alpha \simeq k + \varepsilon$. **Right:** Feature overlap between the learned first-layer representations $\{\hat{h}_\mu^{(1)}\}_{\mu=1}^n$ and the ground truth (see Appendix B for details). Consistently with the behaviour of the MSE, the overlap increases significantly beyond the same threshold, illustrating the robustness of the hierarchical estimator to this choice of non-linearity.

A natural performance measure for $\widehat{A}^{(1)}$ is therefore an *overlap*, which is plotted in the central panels of the figures in the main text, that measures similarity in terms of Frobenius norm $(\|\cdot\|_F)$

$$q_A^{(1)} = \|A^{(1)}(\widehat{A}^{(1)})^\top\|_F \tag{43}$$

During training, this rotation propagates through the model, so that at the population level we have $\widehat{\mathbf{h}}^{(1)} = R\mathbf{h}^{(1)}$. Plugging in this expression into $\hat{C}_2^{(2)}$ eq. (21) and recalling that the Hermite are covariant with rotations $(H_2(R\mathbf{x}) = RH_2(\mathbf{x})R^\top)$, the next moment matrix then satisfies

$$\widehat{C}_2^{(2)} = \frac{1}{n}\sum_{\mu=1}^n y_\mu H_2(\widehat{\mathbf{h}}_\mu^{(1)}) = R\left(\frac{1}{n}\sum_{\mu=1}^n y_\mu H_2(\mathbf{h}_\mu^{(1)})\right)R^\top, \tag{44}$$

and therefore estimates the matrix $RA^{(2)}R^\top$ rather than $A^{(2)}$ itself. While, no direct overlap is convenient for this estimator, the rotation vanishes when computing the latent feature $h^{(2)}$ and thus after defining a test dataset, an MSE on $y$ becomes a good measure to test the performance of the model.

**Latent features overlap.** We consider a test dataset $\{(\mathbf{x}_\mu, y_\mu)\}_{\mu=1}^{n_{\text{test}}}$. Using this dataset, we evaluate the estimators $\widehat{\mathbf{h}}^{(1)}$ of the latent feature $\mathbf{h}^{(1)}$. Let us call the feature matrix $\hat{H}^{(1)} = \{\hat{\mathbf{h}}_\mu^{(1)}\}_{\mu=1}^{n_{\text{test}}} \in \mathbb{R}^{n_{\text{test}} \times \hat{d}^\varepsilon}$. We introduce the feature overlap with the ground truth features $H^{(1)} \in \mathbb{R}^{n_{\text{test}} \times d^\varepsilon}$

$$q_\mathbf{h}^{(1)} = \|\mathbf{h}^{(1)}(\widehat{\mathbf{h}}^{(1)})^\top\|_F^2, \tag{45}$$

which cancels out the rotation $R$ arising in the estimation of $A^{(1)}$ (see eq. (44)). One can further observe that the rotation $R$ vanishes asymptotically in the computation of the second-layer feature. Indeed, at the population level,

$$\widehat{h}_\mu^{(2)} = \langle\widehat{A}^{(2)}, H_2(\widehat{\mathbf{h}}_\mu^{(1)})\rangle_F = \langle RA^{(2)}R^T, RH_2(\mathbf{h}_\mu^{(1)})R^T\rangle_F = h_\mu^{(2)} \tag{46}$$

The panels showing the feature overlap in the main figures have been generated averaging over 10 different seeds, with error bars given by standard deviation.

**Computing the MSE.** At training time, once the latent features are estimated $\{\widehat{\mathbf{h}}_\mu^{(1)}, \widehat{h}_\mu^{(2)}\}_{\mu=1}^n$ we produce estimates for the labels $\hat{y}_\mu$ by performing standard kernel regression. Calling the effective preprocessed dataset at hand $\mathcal{D}_2 = \{\widehat{h}_\mu^{(2)}, y_\mu\}_{\mu=1}^n$ we construct the empirical risk:

$$\mathcal{R}(\mathbf{a}; \mathcal{D}_2) = \sum_{\mu=1}^n \left(y_\mu - \langle\mathbf{a}, \varphi(\widehat{h}_\mu^{(2)})\rangle\right)^2 + \lambda\|\mathbf{a}\|_2 \tag{47}$$

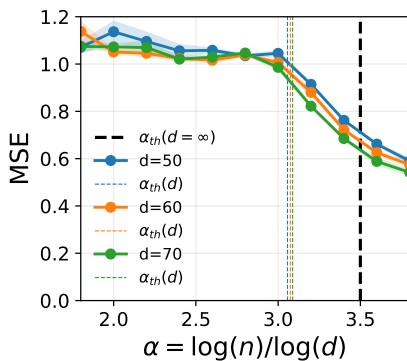 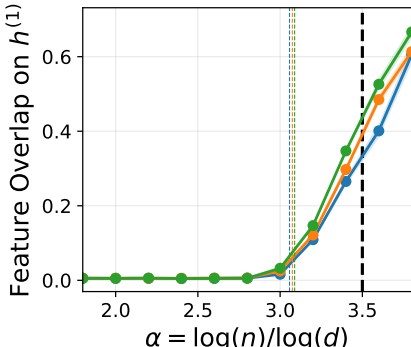

*Figure 6.* **Learning with a third-order Hermite polynomial** ($k = 3$). Performance of the hierarchical estimator described in Algorithm 1 when learning the target (10) with a third-order Hermite polynomial in the first layer. **Left:** Mean Squared Error (MSE) of the label predictor $\{\hat{y}_\mu\}_{\mu=1}^n$ as a function of the normalized number of samples $\alpha = \log(n)/\log(d)$, for different input dimensions $d = \{50, 60, 70\}$. The latent feature dimension is fixed to $d^\varepsilon = \sqrt{d}$. Vertical colored dashed lines indicate the finite-size theoretical thresholds $\alpha_{\text{th}}(d)$ accounting for the effective dimension of the symmetric third-order feature space, while the black dashed line corresponds to the asymptotic prediction $\alpha = k + \varepsilon$. **Right:** Feature overlap between the learned first-layer representations $\{\hat{h}_\mu^{(1)}\}_{\mu=1}^n$ and the ground truth (see Appendix B). Both the MSE and the feature overlap exhibit a clear transition around the finite-size thresholds, with stronger finite-size effects compared to the $k = 2$ case.

where $\mathbf{a} \in \mathbb{R}^p$ is the predictor in the $p-$ dimensional feature space defined by the feature map $\varphi : \mathbb{R} \to \mathbb{R}^p$. By minimizing this empirical risk, we thus obtain our estimates $\hat{y}(h) = \langle \hat{\mathbf{a}}, \varphi(h) \rangle$. The figures in the main are produced using a polynomial kernel estimator of degree 7 where the regularization parameter is tuned with cross validation over a fixed range $\{10^{-3}, 10^{-4}, 10^{-5}\}$.

Thus, we evaluate the learned labels estimate by considering the Mean Squared Error (MSE) over a fresh test dataset $\mathcal{D}_{\text{test}}$:

$$\text{MSE} = \frac{1}{n_{\text{test}}} \sum_{\mu=1}^{n_{\text{test}}} (\hat{y}_\mu - y_\mu)^2 \tag{48}$$

Further, we consider the normalized MSE by dividing by the variance of $y_\mu$. The panels showing the feature overlap in the main figures have been generated averaging over 10 different seeds, with error bars given by standard deviation.

**Varying the non-linearity.** In addition to the results shown in the main text, we also consider values of $g^\star$ different from the identity (Fig. 2) and $\tanh$ (Fig. 4). In Fig. 5, we present results for a centered ReLU non-linearity, $g^\star(x) = \text{relu}(x) - \mathbb{E}_{\xi \sim \mathcal{N}(0,1)}[\text{relu}(\xi)]$, illustrating that other functions satisfying the assumptions on $g^\star$ can also be handled by the method.

**Relaxing the Hermite order of the $1^{\text{st}}$ layer.** Another setting of interest consists in relaxing the condition regarding the order of the Hermite polynomials considered, being it always $k = 2$ in the main text. Although going to higher $k$ presents numerical challenges with $\mathcal{O}(d^k)$ scaling of the effective number of parameters, in Fig. 6, we report numerical results obtained with a third-order Hermite polynomial ($k = 3$) in the first layer.

For $k = 3$, the effective number of parameters is given by the dimension of the symmetric third-order tensor space, $D_1 = \mathcal{B}(d, 3) = d(d+1)(d+2)/6$. Although this quantity is asymptotically equivalent to $d^3$ in the high-dimensional limit, finite-size prefactors are numerically significant at the dimensions accessible in practice. As a consequence, finite-size effects cannot be neglected when interpreting the location of the learning transition. We therefore define the effective sample complexity threshold as

$$\alpha_{\text{th}}(d) = \log_d \left( \frac{d(d+1)(d+2)}{6} \right) + \varepsilon, \tag{49}$$

which converges to $3 + \varepsilon$ only in the asymptotic limit $d \to \infty$. The thresholds reported in Fig. 6 account for this finite-size correction and show good agreement with the numerical results.

**Possibility of Backward Feature Correction.** Consider the simplest example with two-fold quadratic compositional target in eq. 10. Computing the expectation of the moment matrix, one can see that the third-layer features $A^{(2)}$ contribute

to the signal as an effective rotation of the eigenvectors (spikes) that will be found with the spectral procedure. This means that although one recovers the features $\{\hat{A}_i\}_{i=1}^{d^\varepsilon}$ up to rotation, once the next layer features $\hat{A}^{(2)}$ are recovered, one can disambiguate and "unrotate" to the original basis to recover the actual $\hat{A}^{(1)}$. This is an example of backward feature correction, as explained first by (Allen-Zhu & Li, 2023).

## C. Spectral learning as implicit gradient descent

The connection between spectral estimation and gradient-based optimization has been extensively studied in recent years. More recently, the development of preconditioned optimization methods (Martens & Grosse, 2015; Kingma & Ba, 2014; Anil et al., 2020) and modern variants such as Shampoo (Gupta et al., 2018; Anil et al., 2020), has further refined our understanding of how curvature information shapes the optimization landscape.

In this section, we make this connection precise for our spectral estimator by showing that gradient descent on a matched architecture implicitly performs power iteration on the moment matrices $\hat{C}_2^{(1)}$ and $\hat{C}_2^{(2)}$. This connection is not entirely new, at least for the estimation of the last layer ($A_2$), and was at the roots of the spectral method discussed in (Lu & Li, 2020; Maillard et al., 2022; Mondelli & Montanari, 2018; Defilippis et al., 2024; Kovačević et al., 2025) and exploited in a variety of papers (e.g. (Bonnaire et al., 2025; Zhang et al., 2025)), albeit with explicit transformation of the target function.

### C.1. Gradient-Based Setting

We consider a matched student-teacher architecture and study the gradient descent updates of the student parameters. We introduce the notation and loss function below, and then compute the corresponding gradients in order to identify the empirical estimators governing the learning dynamics.

Throughout this analysis, we focus on the early-time training dynamics starting from small random initialization. In this regime, the parameters remain close to their initial values and the gradients are dominated by terms that are linear in the current estimates. As a consequence, the learning dynamics are driven by empirical moment matrices, while higher-order interaction terms can be neglected at leading order.

We write the teacher with flattened version of the first Hermite polynomial, namely we note $\phi_\mu = F[H_k(\mathbf{x}_\mu)] \in \mathbb{R}^{d^k}$, also $F[\{A_i\}_{i=1}^{d^\varepsilon}] = A^{(1)} \in \mathbb{R}^{d^\varepsilon \times d^k}$. Then, one can write the teacher as:

$$y_\mu = \langle A^{(2)}, (A^{(1)}\phi_\mu)(A^{(1)}\phi_\mu)^T - I_p \rangle. \tag{50}$$

Then we define a fitting model whose architecture matches precisely the target one. Thus, by noting the estimators at a time $t \geq 0$, $\hat{A}_t^{(1)} \in \mathbb{R}^{d^\varepsilon \times d^k}$ and $\hat{A}_t^{(2)}$, one can define the estimator on the label:

$$\hat{y}_t \equiv \hat{y}_t(\hat{A}_t^{(1)}, \hat{A}_t^{(2)}) := \langle \hat{A}^{(2)}, (\hat{A}^{(1)}\phi_\mu)(\hat{A}^{(1)}\phi_\mu)^T - I_p \rangle. \tag{51}$$

We consider a standard squared loss for the matched architecture,

$$\mathcal{L} \equiv \mathcal{L}(A^{(1)}, A^{(2)}) = \frac{1}{2n} \sum_\mu \left( y_\mu - \hat{y}_\mu(A^{(1)}, A^{(2)}) \right)^2. \tag{52}$$

We consider the Gradient Descent procedure to update the latent estimators $\hat{A}_t^{(1)}$ and $\hat{A}_t^{(2)}$, it reads:

$$\forall i \in [p], \ \hat{A}_{i,t+1}^{(1)} = \hat{A}_{i,t}^{(1)} - \eta \boldsymbol{\nabla}_{A_i^{(1)}} \mathcal{L}(\hat{A}_t^{(1)}, \hat{A}_t^{(2)}), \tag{53}$$

$$\forall i \in [p], \ \hat{A}_{i,t+1}^{(2)} = \hat{A}_{i,t}^{(2)} - \eta \boldsymbol{\nabla}_{A_i^{(2)}} \mathcal{L}(\hat{A}_t^{(1)}, \hat{A}_t^{(2)}). \tag{54}$$

## C.2. Emergence of the first-layer estimator

Let us consider that the estimator at the second layer $\widehat{A}_t^{(2)}$ are fixed and non vanishing. Then, for the first layer, the gradient of the loss reads:

$$\boldsymbol{\nabla}_{A_i^{(1)}} \mathcal{L} = \frac{-2}{n} \sum_\mu (y_\mu - \widehat{y}_\mu)\, (\widehat{\mathbf{A}}_{i,t}^{(2)})^T \widehat{A}_t^{(1)} \left( H_2(\phi_\mu) + I_{d^k} \right) \tag{55}$$

$$= -2(\widehat{\mathbf{A}}_{i,t}^{(2)})^T \widehat{A}_t^{(1)} \left( \widehat{C}_k^{(1)} + I_{d^k} \right) + (\widehat{\mathbf{A}}_{i,t}^{(2)})^T \widehat{A}_t^{(1)} \left( \frac{2}{n} \sum_\mu \widehat{y}_\mu \left( H_2(\phi_\mu) + I_{d^k} \right) \right). \tag{56}$$

The above expression shows that, at leading order, the gradient update for the first-layer parameters involves the empirical moment matrix $\widehat{C}_k^{(1)}$ introduced in the main text. In the early-time regime, this matrix therefore governs the learning dynamics of $\widehat{A}_t^{(1)}$. The second order computation leads to:

$$\mathcal{H}_{ij} = \boldsymbol{\nabla}_{A_j^{(1)}} \boldsymbol{\nabla}_{A_i^{(1)}}^T \mathcal{L} = -2\widehat{A}_{ij}^{(2)} (\widehat{C}_k^{(1)} + I_{d^k}) + \widehat{A}_{ij}^{(2)} \left( \frac{2}{n} \sum_\mu \widehat{y}_\mu \left( H_2(\phi_\mu) + I_{d^k} \right) \right) \tag{57}$$

$$+ (\widehat{\mathbf{A}}_{i,t}^{(2)})^T \widehat{A}_t^{(1)} \left( \frac{2}{n} \sum_\mu \left( H_2(\phi_\mu) + I_{d^k} \right)^2 \right) (\widehat{A}_t^{(1)})^T \widehat{\mathbf{A}}_{j,t}^{(2)}. \tag{58}$$

Thus, at initialization ($\widehat{A}_{t=0}^{(1)} \simeq 0$), it yields:

$$\mathcal{H}_{t=0} \simeq \widehat{A}_{t=0}^{(2)} \left( \widehat{C}_k^{(1)} + I_{d^k} \right). \tag{59}$$

As a result, the initial evolution of the first-layer parameters is dominated by a linear action of $\widehat{C}_k^{(1)}$, leading to an implicit power-iteration–like behavior that progressively aligns the estimates with its leading eigenspaces.

## C.3. Gradient structure for the second layer

The computation of the gradient for the second layer is straightforward and yields:

$$\nabla_{A^{(2)}} \mathcal{L}_t = \frac{-1}{n} \sum_\mu (y_\mu - \widehat{y}_\mu) H_2(\widehat{\mathbf{h}}_\mu^{(1)}) = -\widehat{C}_2^{(2)} + \frac{1}{n} \sum_\mu \widehat{y}_\mu H_2(\widehat{\mathbf{h}}_\mu^{(1)}). \tag{60}$$

Thus at initialisation with $\widehat{A}_{t=0}^{(2)} \simeq 0$, the gradient reads:

$$\nabla_{A^{(2)}} \mathcal{L}_{t=0} = -\widehat{C}_2^{(2)}. \tag{61}$$

This shows that, at initialization, gradient descent on the second-layer parameters is directly driven by the empirical matrix $\widehat{C}_2^{(2)}$, whose leading eigenspaces define the spectral estimator studied in the main text.

## C.4. Conclusion

Importantly, the empirical matrices $\widehat{C}_k^{(1)}$ and $\widehat{C}_2^{(2)}$ that appear naturally in the gradient dynamics are exactly the estimators used by our spectral procedure. While the spectral method explicitly constructs these matrices and extracts their leading eigenspaces, gradient descent implicitly accesses the same objects through its update rules. This explains why both approaches exhibit the same learning thresholds and recovery behavior.

