# OpenReview forum: "Deep Learning of Compositional Targets with Hierarchical Spectral Methods"
_ICML.cc/2026/Conference — ICML 2026 regular_

### Official Review · Reviewer_M99F · 2026-02-13

**Soundness:** 3
**Presentation:** 2
**Significance:** 3
**Originality:** 3
**Overall Recommendation:** 5
**Confidence:** 5

**Summary:**

The paper studies when depth provides a genuine computational and statistical advantage in a controlled high-dimensional Gaussian model. The target function is globally a high-degree polynomial, but it has a compositional structure: it factors through simpler intermediate features.
Instead of analyzing gradient descent, the authors propose an explicit layer-wise spectral algorithm. The method constructs suitable Hermite moment matrices and recover the latent subspaces sequentially, one layer at a time. The main results characterize the sample sizes required for each stage to succeed. This leads to an overall sample complexity d^k. The analysis is based on Gaussian equivalence argument.

**Compliance With Llm Reviewing Policy:**

Affirmed.

**Final Justification:**

I have raised the score to 5 since the concerns are addressed.

**Key Questions For Authors:**

Please see the weakness part.

**Limitations:**

Please see the weakness part.

**Strengths And Weaknesses:**

This paper is quite interesting and provides significant insight into hierarchical feature learning. I am a domain expert in this area, and I am impressed by the technical results.

That said, there are several framing issues that make the paper unsuitable for publication in its current form. I am not suggesting that the work itself is weak. The results and insights are strong. However, the way the paper positions and discusses prior work needs improvement. These issues should be straightforward to fix. If they are properly addressed, I would definitely raise my score to 5 (clear accept).

1. Improper discussion of related work. For instance, the paper states: “Closer to us, and a major inspiration to the present work, is the series of papers (Wang et al., 2023; Nichani et al., 2024; Fu et al., 2025) who discussed compositional functions and demonstrated the advantage of three-layer nets over two layers and kernel methods.” The ordering is incorrect: Nichani et al., 2024 precedes Wang et al., 2023 in this line, despite appearing later (please check the arXiv for the correct time order). More importantly, the comparison is inaccurate. For example, Wang et al., 2023 studies the general case with arbitrary k, while Nichani et al., 2024 focuses on the specific case k = 2. The paper states the opposite, which is incorrect.

Furthermore, consider the statement “For the same example, (Fu et al., 2025) shows that a three-layer network can …”. The setting in Fu et al., 2025 cannot be recovered as a special case of the present submission. In Fu et al., 2025, the number of quadratic features is a nonzero constant (Therefore the gaussian equivalence for that space fails. In the present submission, since d^{\epsilon} is still large enough, we still have gaussian equivalence for that). In addition, in Equation (3) of this submission, only the second degree feature is considered. In Fu et al., 2025 the relevant feature for this layer is linear rather than quadratic. These two cases are fundamentally different.

2. Insufficient discussion of related work. The paper misses some prior works on learning multi-index models (e.g., [https://arxiv.org/abs/2205.01445](https://arxiv.org/abs/2205.01445) and [https://arxiv.org/abs/2602.01434](https://arxiv.org/abs/2602.01434)), as well as related results on hierarchical learning (e.g., [https://arxiv.org/abs/2304.01063](https://arxiv.org/abs/2304.01063)).

---

> ### Author Rebuttal · Authors · 2026-03-30
>
> We thank the reviewer for the positive assessment of the technical results and for the constructive comments on framing and related work.
>
> We address the two weaknesses below.
>
> 1. Regarding "Improper discussion of related work", we agree that some comparisons in the current version are not stated accurately enough. In particular, we will correct the chronological ordering and sharpen the description of the scope of the papers by Nichani et al. and Wang et al. More specifically, Wang et al. study a more general degree-$k$ setting, while Nichani et al. focus on a more specific quadratic-feature regime.
>
>     We also agree that our current wording around Fu et al. should be sharpened. Our intention was not to claim that the setting of Fu et al. is a special case of our model, nor that our algorithm should be read as "beating" theirs. The two settings differ in important ways, and these differences do affect the corresponding behavior and analysis.
>
>     That being said, we still believe it is useful to discuss these works together for non-expert readers, because they belong to the same broader line of research on hierarchical feature learning. A better way to phrase our point is that, if one wants to move from the type of setting studied in Fu et al. toward genuinely deeper compositional hierarchies, the type of model we study is a natural first step, even though it lies in a different regime and requires different techniques. We will revise this discussion accordingly and make both the similarities and the differences explicit.
>
>
> 2. Regarding "Insufficient discussion of related work", we thank the reviewer for pointing out the missing references. We will incorporate these works in the revised version, in particular the papers mentioned by the reviewer on multi-index models and hierarchical learning, and clarify their relation to our contribution.

---

> > ### Author Rebuttal · Reviewer_M99F · 2026-04-02
> >
> > Thanks for the rebuttal. My concerns have been addressed.

---

### Official Review · Reviewer_4EiF · 2026-03-11

**Soundness:** 3
**Presentation:** 3
**Significance:** 2
**Originality:** 2
**Overall Recommendation:** 4
**Confidence:** 4

**Summary:**

The current work explores the role of depth in deep neural networks. As explained by the authors, the sample complexity aspects of depth have been studied through expressibility considerations, or the context of deep linear networks, and in layer-wise training protocols. The authors offer an alternative to layer-wise GD, via a spectral learning algorithm. The general idea is to compute covariances matrix of features and determine the sample complexity when an outlier emerges from the bulk. Using this training protocol, they prove that such targets can be learn at virtually optimal sample complexity (from an information theory perspective).

**Compliance With Llm Reviewing Policy:**

Affirmed.

**Final Justification:**

see discussion

**Key Questions For Authors:**

To what extent can the authors relate their work to gradient flow, online learning, SGD, Bayesian Neural Networks, or kernel-based approaches?

**Limitations:**

See weaknesses

**Strengths And Weaknesses:**

Strengths: The authors address a timely and interesting problem, which is that, indeed, theory of deep learning is to some extent too shallow at the moment [typical solvable models have one or two trainable layers and sequential layer-wise extensions]. The algorithm that they suggest plausibly leads to the same sample complexity as standard training.

Weaknesses: The main weakness of the work is that it somewhat circumvents the challenges it seeks to address. Indeed, practical experience, expressibility results, results on layer-wise training all suggest that depth is useful for its ability to capture hierarchies beyond interpolations. An outstanding issue is how one studies the implicit bias of neural networks trained using common algorithms. The authors suggest a very different training procedure, which is much less practical for wide neural networks, and so the contribution is somewhat tangential.

Another weakness concerns the literature review. The authors don’t cite literature on student-teacher settings (https://arxiv.org/pdf/2402.06323), Bayes-Optimal teacher-student settings (https://arxiv.org/pdf/2510.24616), which also study sample complexity of (Bayesian) deep neural networks. Similarly, DMFT, kernel adaptations, and rainbow networks approaches also focus on spectral aspects of learning, where spectral outliers similarly reflect learning.

---

> ### Author Rebuttal · Authors · 2026-03-30
>
> We thank the reviewer for the positive assessment and for the constructive comments.
>
> Regarding the concern that the proposed procedure
>
> > "somewhat circumvents the challenges it seeks to address"
>
> and the related question:
>
> > "To what extent can the authors relate their work to gradient flow, online learning, SGD, Bayesian Neural Networks, or kernel-based approaches?"
>
> Our goal is not to propose a disconnected training paradigm, but rather a controlled and analyzable surrogate for layer-wise gradient-based learning. This is precisely why, instead of analyzing gradient descent directly, we introduce an explicit hierarchical spectral procedure that isolates the same progressive feature-learning mechanism in a tractable form.
>
> More specifically, in Appendix C we already make this connection explicit in the matched teacher-student setting considered in the paper: at small initialization and in the early stages of full-batch gradient descent, the updates are governed by the same empirical moment matrices as those used by our spectral estimator. In that sense, the spectral method should be understood as a proxy for the gradient-based mechanism we want to study, rather than as a tangential alternative.
>
> Regarding the broader frameworks mentioned by the reviewer, we view them mainly as relevant points of contact for positioning the paper rather than as direct objects of comparison for the present analysis. We will therefore expand the related-work discussion accordingly, and incorporate the references pointed out by the reviewer, in particular on student-teacher and Bayes-optimal settings, as well as other spectral perspectives on learning.

---

> > ### Author Rebuttal · Reviewer_4EiF · 2026-04-04
> >
> > I thank the authors for pointing out Appendix C, which indeed fleshes out connections to standard training under some restricting assumptions. I raise my score to weak accept.

---

> > > ### Author Response · Authors · 2026-04-06
> > >
> > > Thank you for the follow-up and for the careful reading.
> > >
> > > We appreciate your clarification and are glad that our responses addressed your concerns. We will incorporate the remaining points you raised into the revised version to improve clarity.
> > >
> > > Thank you again for your helpful feedback.

---

### Official Review · Reviewer_pXHD · 2026-03-11

**Soundness:** 2
**Presentation:** 1
**Significance:** 2
**Originality:** 2
**Overall Recommendation:** 2
**Confidence:** 2

**Summary:**

The paper studies whether hierarchical (multi-stage) learning procedures can learn certain compositional target functions more efficiently than shallow methods. In a high-dimensional Gaussian setting, the authors propose and analyze a staged spectral algorithm that recovers intermediate representations sequentially and show that this approach can achieve lower sample complexity than kernel or shallow estimators for the constructed targets.

**Compliance With Llm Reviewing Policy:**

Affirmed.

**Final Justification:**

The clarifications on the scope of the paper, the partial agnosticism of the first-layer procedure, and the planned presentation improvements are helpful.

However, my main concerns remain. The claimed advantage appears closely tied to the match between the hierarchical target class and the estimator, so it remains unclear how much should be attributed to depth itself rather than to this specific (co-)designed setting. The connection to broader deep architectures remains limited.

Overall, the rebuttal was helpful but did not change my assessment. My recommendation therefore remains unchanged.

**Key Questions For Authors:**

- The paper is motivated by the empirical efficacy of depth, but the analysis and experiments focus only on a very limited low-depth hierarchical setting. To what extent do the authors expect the main conclusions to extend to genuinely deeper architectures, rather than this specific three-stage construction?

- More broadly, how much of the observed advantage is a consequence of depth itself, versus the fact that the proposed hierarchical spectral method is explicitly matched to the compositional structure of the target distribution?

**Limitations:**

yes

**Strengths And Weaknesses:**

## Strengths

- The computational advantage of depth is an important and highly relevant theoretical question.
- The connection discussed in the appendix section *“Spectral learning as implicit gradient descent”* is interesting and appears to be a promising direction for further investigation.

## Weaknesses

- The presentation is currently difficult to parse, which makes it very challenging to evaluate and appreciate the technical contributions.
- More substantively, the theoretical setup appears overly curated: the authors define a particular hierarchical targetclass and then analyze an algorithm explicitly designed to exploit that very same structure. This raises the concern that the separation may be driven largely by the co-design of the data model and estimator rather than by a robust or broadly meaningful advantage of depth.
- While the paper is motivated by the general question of the advantage of depth, the analysis and experiments are confined to a very specific controlled setting with a shallow compositional hierarchy. The connection between this setup and the broader motivation is therefore somewhat limited.
- There are several exposition issues already in Section 2. For example, $H_1$ and $H_2$ are introduced without sufficient context, and key assumptions are not explained or justified (e.g., the claim on lines 105–106 that the outer nonlinearity can be assumed to be a polynomial of degree $p$).
- The subsection titled *“Expected performance: informal discussion”* is misleadingly named: it is neither especially informal nor really a discussion of expected performance, but instead presents scaling heuristics for asymptotic sample complexity.
- Key concepts are introduced without sufficient explanation. For example, the discussion of limitations of kernel methods and random feature models appears before these notions are defined or intuitively explained, making the argument difficult to follow for readers unfamiliar with this literature.
- The contributions are not presented clearly enough; and it is often difficult to distinguish between what is rigorously proved, what is heuristic, and what is supported only empirically.
- The statement that “a large part of the theory of neural networks comes from regimes where features are effectively fixed during training” (line 146) is too broad and somewhat misleading. Kernel and random feature regimes are important and mathematically tractable, but they represent only one strand of the broader theory literature.

## Minor comments

- Line 244: punctuation/typo in “i.e though”.
- The heading *“Conclusions, discussions, limitations”* is stylistically awkward; a title such as *“Discussion and Limitations”* would be clearer.

---

> ### Author Rebuttal · Authors · 2026-03-30
>
> We thank the reviewer for acknowledging our analysis, in particular the role of depth in the learning strategy and the data model, as well as its connection to gradient descent.
>
> We first address the key questions:
>
> - **Key Question 1:**
>   In short, we expect the same behavior to extend to deeper models. First, preliminary numerical simulations support this claim, although presenting them would require additional modeling details that would detract from the clarity of the current manuscript. Second, from a theoretical perspective, we expect Gaussian equivalence arguments to extend across layers, leading to similar conclusions in deeper settings. We will emphasize this in the revised version.
>
> - **Key Question 2**:
>     We do not fully agree with the characterization that the observed separation is merely a consequence of a fully teacher-student co-designed setting. At the first layer, our procedure is already partially agnostic: the algorithm scans candidate polynomial degrees and detects low-rank structure directly from data, rather than assuming the first-layer latent subspace in advance (see Algorithm 1). In that sense, the estimator is not given the teacher parameters, but only a hierarchical inductive bias. That said, we agree that this agnosticism is only partial. In the current analysis, the second layer is indeed specialized to an order-2 Hermite estimator, and therefore still benefits from prior structural knowledge on the target class. We will make this limitation more explicit in the revised version. Extending the method to more agnostic higher-layer procedures is a non-trivial and important direction for future work. This is also consistent with the discussion in the conclusion, where we explicitly mention moving beyond the present Hermite-based setting as a main open direction.
>
>     A natural next step is to formalize this intuition within multi-layer random-feature models. With sufficiently many random features at each layer, one may hope to obtain a more agnostic hierarchical procedure, where the relevant low-degree components are selected from data rather than hard-coded in the estimator. We do not analyze such models in the present paper, but we believe this is an important direction for future work, precisely because it could help disentangle which part of the gain comes from depth itself and which part comes from the specific estimator considered here.
>
>     More fundamentally, the main point established in the manuscript is that the gain comes from the ability to exploit compositional structure through intermediate representations. Shallow kernel methods depend essentially on the overall polynomial degree of the target and are therefore insensitive to its hierarchical organization, whereas our hierarchical procedure turns the global high-degree problem into a sequence of lower-order estimation tasks. This is the precise sense in which the observed gain should be attributed to depth, rather than to exact knowledge of the teacher parameters.
>
> We address the weaknesses in three points.
>
> - **Presentation and readability.** We agree that the presentation can be improved. In the revised version, we will streamline the narrative in Section 2, introduce the latent features $h^{(1)}$ and $h^{(2)}$ with more context, add brief background on kernel and random-feature methods before discussing their limitations, and distinguish more clearly between rigorous results, heuristics, and numerical evidence. We will also rename the subsection "Expected performance: informal discussion" to a more accurate title.
>
> - **Technical clarifications.** We thank the reviewer for pointing out that some assumptions require better motivation. In particular, the assumption that $g^\star$ is polynomial is mainly used for the proof; numerically, we also consider non-polynomial choices such as $\tanh$ and centered ReLU. More broadly, our analysis relies on the standard viewpoint that nonlinearities can be studied through their Hermite expansion, with the relevant information carried by the low-degree terms. We will clarify this point earlier and more explicitly. Concerning the distinction between rigorous and empirical claims, in the conclusion we explicitly note that the proof is restricted to $\varepsilon < 1/2$, while the broader validity beyond this regime is only supported numerically. If the reviewer has more specific passages in mind, we would be happy to clarify them in the revised version.
>
> - **Wording issues.** We agree that the sentence "a large part of the theory of neural networks comes from regimes where features are effectively fixed during training" is too broad. Our point was simply to refer to an important and mathematically tractable strand of the literature, namely fixed-feature regimes such as kernel and random-feature models. We will revise the sentence accordingly.
>
> Regarding the two minor comments, we thank the reviewer for pointing them out and will update the manuscript accordingly.

---

> > ### Author Rebuttal · Reviewer_pXHD · 2026-04-02
> >
> > Thank you for the rebuttal. While the clarifications are helpful, my main concerns remain. In particular regarding whether the gains stem from depth itself or from alignment with the target structure, and the limited connection to broader deep architectures. Therefore, my assessment remains unchanged.

---

### Official Review · Reviewer_nR5H · 2026-03-13

**Soundness:** 2
**Presentation:** 3
**Significance:** 2
**Originality:** 3
**Overall Recommendation:** 4
**Confidence:** 3

**Summary:**

This paper studies how depth reduces sample complexity for learning structured functions in high dimensions. The target is a compositional polynomial built through two latent stages where the first layer is a $d^\epsilon$-dimensional function constructed using order-$k$ Hermite tensors and the second layer is from order-2 Hermite. The paper proposes to use a two-stage method that uses the Hermite moment matrices and PCA-type spectral methods. The paper argues that this staged recovery exploits compositional structure that shallow methods cannot use directly. The main claim is that the first latent layer can be recovered when $n \gg d^{k+\epsilon}$ and the second latent scalar feature can then be recovered with $n \gg d^{2 \epsilon}$. This is presented as a sharp separation from shallow approaches. The theory is built on a Gaussian equivalence principle for Hermite features, and simulations studies are conducted.

**Compliance With Llm Reviewing Policy:**

Affirmed.

**Final Justification:**

The rebuttal and follow-up responses have substantially clarified my main technical concerns, especially regarding the derivation of equations (30) and (35). The proposed route now appears plausible to me. I therefore raised my evaluation.

**Key Questions For Authors:**

1. Can the authors provide a quantitative refinement of Lemma 4.2 that yields the $O\!\left(1/\sqrt{d_1}\right)$ rate used in the final step of Theorem 3.3, rather than only an $o(1)$ bound under the same scaling?

2. Can the authors provide a complete derivation of the Matrix Bernstein argument used to establish equations (29) and (33), including the bounds on the centered summands and the relevant variance proxy?

3. For equations (30) and (35), can the authors explicitly prove the Lipschitz bound that is uniform over all $\|v\|_2=\sqrt{d_1}$, so that Lemma A.1 indeed yields an operator norm bound?

4. In the construction of the estimator, it seems that the identity


$$A^{(1)}\big((A^{(1)})^\top A^{(2)} A^{(1)}\big)(A^{(1)})^\top = A^{(2)}$$

is being used. Does the model assume that

$$A^{(1)}(A^{(1)})^\top = I_{d_1}?$$

If not, could the authors explain why this does not affect the estimator or the proof?

5. Can the authors provide numerical evidence for the remark following Theorem 3.5, namely that the same error bounds should also hold when the practical estimator based on $\hat{\mathbf h}^{(1)}_\mu$ is used instead of the oracle estimator based on the true $\mathbf h^{(1)}_\mu$?


A satisfactory resolution of Questions 1 to 3 would materially improve my assessment of the paper’s technical soundness.

**Limitations:**

yes

**Strengths And Weaknesses:**

**Significance**

The problem is important, and the paper proposes a clean mechanism for a depth advantage. The claimed scaling $O(d^{k+\epsilon})$, compared with a shallow dependence on the overall degree, would be a meaningful theoretical separation if fully justified. However, the connection to gradient descent remains indirect, and it is still unclear how this framework could lead to a rigorous theory for standard deep networks.

**Originality**

The hierarchical spectral learning framework is interesting. Decomposing the target into sequential low order recovery steps provides a novel perspective, and extending Gaussian equivalence to this hierarchical setting also appears new.

**Presentation**

The paper is reasonably well presented. It has a clear top level narrative, and the main theoretical developments are stated explicitly.

**Soundness**

The Gaussian and Hermite setting is narrow, but it is analytically purposeful and makes the problem mathematically tractable. The simulations are aligned with the theoretical message.

The paper has a strong theoretical core, but the current version does not yet fully establish the advertised guarantees. Theorem 3.5 is a result on the idealized estimator, rather than the actual estimator.

Below are the main gaps I found in the proofs of Theorems 3.3 and 3.5.

1. The concentration argument leading to equations (29) and (33) is too terse and should be derived more carefully. In particular, applying Matrix Bernstein requires explicit control of the centered summands and the corresponding variance proxy. The current proof omits these steps. This issue may be fixable, but it should be written out.
2. The final step in the proof applies Lemma 4.2 in a way that does not match the claimed rate. As stated, Lemma 4.2 appears to yield an $o(1)$ bound after the same $\sqrt{d_1}$ scaling, not the stronger $O(1/\sqrt{d_1})$ rate used in Theorem 3.3. Unless there is an additional quantitative refinement of Lemma 4.2 that is not stated, the advertised final term is not proved. The same issue appears in the proof of Theorem 3.5.

---

> ### Author Rebuttal · Authors · 2026-03-30
>
> We thank the reviewer for the careful reading and for the constructive technical comments. We agree that addressing these points more explicitly would improve the paper. We answer the five questions in order below.
>
> 1. We thank the reviewer for pointing this out. We indeed use a quantitative version of Lemma 4.2 that yields the $O(1/\sqrt{d_1})$ bound. The operator norm of $A^{(2)}$ is $O(1/\sqrt{d_1})$, while the Gaussian approximation of $h_2$ yields another factor of $O(1/\sqrt{d_1})$. We will update Lemma 4.2 to make this quantitative refinement explicit.
>
> 2. The Matrix Bernstein bound is applied to the truncated rank-one components in Eq.~(28), which have bounded operator norm. We will provide the full derivation in the revised version, including the bounds on the centered summands and the corresponding variance proxy, following Wen et al. 2025.
>
> 3. The labels $y$ as a function of $h_1$ are given by the mapping $h_1 \mapsto g^\star(h_1^\top A^{(2)} h_1)$. Hence, the required uniform Lipschitz control follows from Assumption 3.2 together with the fact that $\|A^{(2)}\|_{\mathrm{op}} = O(1/\sqrt{d_1})$ with high probability. We will include the full argument in the revised version.
>
> 4. No exact identity $A^{(1)}(A^{(1)})^T=I_{d_1}$ is assumed. What is used is that, under Assumption 3.1, the rows of $A^{(1)}$ are asymptotically orthonormal, so that $A^{(1)}(A^{(1)})^T$ concentrates around $I_{d_1}$ in the regime considered. We will clarify this point in the main text and in the proof.
>
> 5. Yes. In fact, the numerical experiments already use the practical estimator based on $\hat h^{(1)}\_\mu$, not the oracle one based on $h^{(1)}_\mu$. Theorem 3.5 is stated for the idealized estimator only because this is the object we currently control rigorously. We will make this distinction clearer in the revised version, so that it is explicit that the algorithm and simulations do not use the teacher latent features.

---

> > ### Author Rebuttal · Reviewer_nR5H · 2026-04-02
> >
> > Thank you for the detailed rebuttal. The clarifications on the Matrix Bernstein step, the practical estimator used in the experiments, and the asymptotic orthonormality of $A^{(1)}$ are helpful. The response regarding Lemma 4.2 is also helpful, but I regard that point as only partially resolved, since the stronger quantitative refinement is not yet shown in the current manuscript.
> >
> > My main remaining concern is the derivation of equations (30) and (35). I may be missing something, but my concern is not whether the map $h^{(1)} \mapsto g^\star\left((h^{(1)})^\top A^{(2)} h^{(1)}\right)$ is Lipschitz. Rather, the issue is the operator-norm bound, that is, the required uniform control over the family of quadratic test functions indexed by $v$, which involves terms of the form
> >  $$\big((v^\top u)^2-1\big)  y(u).$$
> >  Thus, even if $y(u)$ is $\widetilde{O}(1)$-Lipschitz, the full test function may still be only $\widetilde{O}(\sqrt{d_1})$-Lipschitz unless one proves something sharper.
> >  For this reason, I still do not see how Lemma A.1, as currently invoked, yields the claimed operator norm bounds in (30) and (35).
> >
> > The clarification on $A^{(1)}A^{(1)\top}$ is helpful as well, but the revised paper should explicitly indicate where asymptotic orthonormality replaces an exact identity and show that the resulting approximation error is negligible at the scale of Theorem 3.3.
> >
> > Overall, the rebuttal addresses part of my concerns, but not enough for me to change my rating at this stage. A convincing clarification of the operator norm step, together with the quantitative refinement related to Lemma 4.2, would improve my assessment of the paper’s technical soundness.

---

> > > ### Author Response · Authors · 2026-04-03
> > >
> > > We thank the reviewer for the follow-up and for clarifying the remaining concern.
> > >
> > > More precisely, after the variational characterization of the operator norm (together with the $\sqrt{d_1}$ normalization used in Theorems 3.3 and 3.5), the relevant family of test functions is indexed by unit vectors $v$ and involves quadratic terms of the form
> > >
> > > $$ u \mapsto \mathrm{He}_2(v^\top u) g^\star(u^\top A^{(2)} u).$$
> > >
> > > The point implicitly used in the proof is that, for every fixed $v$ with $||v||=1$, one has $v^\top h^{(1)}=\mathcal O(1)$ with high probability. Conditioning on this event, $\mathrm{He}_2(v^\top h^{(1)})$ and its derivative are uniformly $\mathcal O(1)$, and Assumption 3.2 together with $||A^{(2)}||\_{\mathrm{op}}=\mathcal O(1/\sqrt{d_1})$ yields a Lipschitz bound that is uniform in $v$. The contribution of the complement of this event is lower-order and can be controlled using polynomial moment bounds on $h^{(1)}$. We will include this full argument explicitly in the revised proof.
> > >
> > > Regarding Lemma 4.2, we also agree that the quantitative refinement should be stated in the manuscript rather than only used implicitly. Writing $B$ for the matrix of Hermite-2 coefficients of the map $u \mapsto g^\star(u^\top A u)$, the refinement we use is
> > > $$ ||B-\nu_1 A||_{\mathrm{op}}=\widetilde{\mathcal O}(1/d).$$
> > > By the variational characterization, this reduces to proving uniformly over $||v||=1$ that
> > >
> > > $$ \mathbb E_{z\sim\mathcal N(0,I_d)} \Big[ g^\star(z^\top A z) \mathrm{He}_2(v^\top z) - \nu_1 (z^\top A z) \mathrm{He}_2(v^\top z) \Big] = \widetilde{\mathcal O}(1/d). $$
> > >
> > > This follows from the same expansion as in the proof of Lemma 2.5 of Wang et al. (2023): the leading contribution is precisely $\nu_1 A$, while all subleading terms are of order $\widetilde{\mathcal O}(1/d)$. We will incorporate this proof in the revised version.
> > >
> > > Finally, we agree that the paper will benefit from making explicit where exact identities are replaced by asymptotic orthonormality, as pointed out by the reviewer.

---

### Decision · Program_Chairs · 2026-04-30

**Decision:**

Accept (regular)

**Comment:**

In this paper, the authors study an idealized analogue of multilayer learning in neural networks,
establish theoretical results, and show their correspondence to simulation results. Based on
the initial reviewer feedback, I think the main concerns about this work fall into
(1) relevance/connection to neural network training, since this is an explicitly stated goal,
(2) mathematical correctness, and (3) writing issues (e.g., distinguishing between what is proved
versus what is believed). Based on the discussion, I think that these concerns are partially addressed
by the authors promised revisions and the connection they point out in Appendix C (which is not
part of the main text). The referees all agree that there is some new conceptual component of this
work compared to prior works on hierarchical learning in neural networks. Based on this, I recommend
the paper is accepted assuming the authors make the promised revisions and incorporate Appendix C better.